# Growth hormone releasing hormone signaling promotes Th17 cell differentiation and autoimmune inflammation

Lin Du[1], Bo Man Ho [1], Linbin Zhou[1], Yolanda Wong Ying Yip[1], Jing Na He[1], Yingying Wei[2], Clement C. Tham [1], Sun On Chan [3], Andrew V. Schally[4,5], Chi Pui Pang[1], Jian Li [1,6] ✉ & Wai Kit Chu [1] ✉

Dysregulation of Th17 cell differentiation and pathogenicity contributes to multiple autoimmune and inflammatory diseases. Previously growth hormone releasing hormone receptor (GHRH-R) deficient mice have been reported to be less susceptible to the induction of experimental autoimmune encephalomyelitis. Here, we show GHRH-R is an important regulator of Th17 cell differentiation in Th17 cell-mediated ocular and neural inflammation. We find that GHRH-R is not expressed in naïve CD4[+] T cells, while its expression is induced throughout Th17 cell differentiation in vitro. Mechanistically, GHRH-R activates the JAK-STAT3 pathway, increases the phosphorylation of STAT3, enhances both non-pathogenic and pathogenic Th17 cell differentiation and promotes the gene expression signatures of pathogenic Th17 cells. Enhancing this signaling by GHRH agonist promotes, while inhibiting this signaling by GHRH antagonist or GHRH-R deficiency reduces, Th17 cell differentiation in vitro and Th17 cell-mediated ocular and neural inflammation in vivo. Thus, GHRH-R signaling functions as a critical factor that regulates Th17 cell differentiation and Th17 cell-mediated autoimmune ocular and neural inflammation.

T helper 17 (Th17) cells are a subset of CD4[+] T cells characterized by the secretion of interleukin-17 (IL-17) and expression of a nuclear transcription factor, retinoic acid receptor-related orphan receptor gamma t (RORγt)[1,2]. Th17 cells regulate the host defense against extracellular pathogens and in the pathogenesis of many autoimmune diseases, including autoimmune uveitis and multiple sclerosis[3]. Signaling from T cell receptor and cytokines IL-6 and transforming growth factor-β (TGF-β) induce the differentiation of naïve T cells into Th17 cells via mediating phosphorylation of STAT3, which is further amplified by signaling from IL-23 and IL-21 in a positive feedback loop[4–7]. The phosphorylation of STAT3 can induce the expression of transcription factors RORγt and RORα that acts as a master switch to regulate the

characteristic set of cytokines IL-17A, IL-17F, IL-21, and IL-22 in Th17 cells[3,8]. However, it has been reported that Th17 cells induced by IL-6 and TGF-β are not sufficient to elicit autoimmune diseases, which require a co-stimulation of IL-23, or another co-stimulation of IL-6, IL-1β, and IL-23 in the absence of TGF-β[9,10]. In addition to these cytokines, however, other factors regulating the differentiation and pathogenicity of Th17 cells remain not very well understood.

Autoimmune uveitis is a sight-threatening disease characterized by a heterogeneous group of intraocular inflammatory disorders. It often occurs with systemic autoimmune syndromes such as ankylosing spondylitis, Behçet's disease and Vogt–Koyanagi–Harada (VKH) disease, in which Th17 cells were considered to be associated with the

[1]Department of Ophthalmology and Visual Sciences, The Chinese University of Hong Kong, Hong Kong, Hong Kong. [2]Department of Statistics, The Chinese University of Hong Kong, Hong Kong, Hong Kong. [3]School of Biomedical Sciences, The Chinese University of Hong Kong, Hong Kong, Hong Kong. [4]Endocrine, Polypeptide, and Cancer Institute, Veterans Affairs Medical Center, Miami, FL, USA. [5]Division of Endocrinology, Department of Medicine, Miller School of Medicine, University of Miami, Miami, FL, USA. [6]Department of Ophthalmology, Affiliated Hangzhou First People's Hospital, Zhejiang University School of Medicine, Hangzhou, China. ✉e-mail: lijian@link.cuhk.edu.hk; waikit@cuhk.edu.hk

disease initiation and development[11–13]. In the experimental auto-immune uveitis (EAU), an animal model mimicking human auto-immune uveitis[14], Th17 cells were found to be a predominantly pathogenic mediator in the development of uveitis by producing a set of proinflammatory cytokines, including IL-17A, IL-17F, IL-22, and GM-CSF[15,16]. Unexpectedly, treating uveitis by neutralizing IL-17A, the major pathogenic cytokine, failed to diminish the disease due to the increase of other Th17-lineage cytokines through a negative feedback loop[17]. These studies highlight the involvement of other potential factors or mechanisms to regulate Th17 cells and their associated proin-flammatory cytokines in autoimmune inflammation.

Growth hormone-releasing hormone (GHRH) is a neurosecretory peptide hormone synthesized by the hypothalamus. Its binding to specific GHRH receptors (GHRH-R) in the pituitary regulates the synthesis of growth hormone (GH). In multiple organs, GH induces the production of insulin-like growth factor 1 (IGF-1), which was recently reported to be a key regulator for Th17-regulatory T (Treg) cell balance in autoimmunity[18–20]. We have demonstrated that GHRH and GHRH-R were expressed in multiple ocular tissues, including the iris, ciliary body, retina, and eye-infiltrating leukocytes in the endotoxin-induced uveitis (EIU) animal model[21]. GHRH signaling was involved in ocular inflammation after lipopolysaccharide (LPS) insults, as the expression of GHRH-R was elevated along with the proinflammatory cytokines of IL-1β and IL-6 in the iris and ciliary body, which can be reduced by the GHRH antagonist, a competitive inhibitor of the receptors[21]. In pre-vious studies, GHRH-R was found in T lymphocytes, which regulate immune response to aging[22]. In addition, GHRH was reported to sti-mulate the secretion of IL-17A from human peripheral blood mono-nuclear cells[23]. In another study, GHRH-R deficient mice were reported to be less susceptible to the induction of experimental autoimmune encephalomyelitis (EAE), a Th17 cell-mediated disease model mimick-ing human multiple sclerosis[24,25]. GH was found to be able to increase T cell proliferation and spleen size in these EAE mice[25]. Interestingly, without exogenous peptide immunization, GHRH-deficient mice did not affect thymic T cell development[26]. In agreement with these results, in cattle, myogenic expression of GHRH led to more abundant T cell lineages of CD2+, CD4+CD25+, and CD4+CD45R+[27]. In humans, administration of the GHRH analog resulted in a significant increase of cells expressing various T cell receptors, while no difference was observed in T cells expressing CD3, CD4, or CD8[28]. In addition, GHRH-R was found in human sarcoidosis granuloma, a Th17 cell-mediated inflammatory disorder resulting in multisystemic granuloma, which can be alleviated by GHRH antagonist with a reduced level of IL-17A in an in vitro granuloma model[29].

In this work, we hypothesize that GHRH-R signaling is involved in the pathogenesis of Th17-mediated autoimmune inflammation through the interaction with Th17 cells and their associated cytokines. Our results demonstrate that GHRH-R is preferentially expressed in Th17 cells, but not Th1, Th2, or Treg cells. GHRH-R signaling activates the JAK-STAT3 pathway, increases the phosphorylation of STAT3, enhances both non-pathogenic and pathogenic Th17 cell differentia-tion and favors the gene expression signatures of pathogenic Th17 cells. The inhibition or deficiency of GHRH-R reduces Th17 cell differ-entiation and alleviates Th17 cell-mediated autoimmune ocular and neural inflammation.

## Results
### Elevated expression of GHRH and GHRH-R in the retina of EAU mice
To investigate the impacts of GHRH and GHRH-R in autoimmune dis-eases, EAU, a well-established autoimmune disease animal model, was used. We first investigated the expression of GHRH and GHRH-R in the retina of EAU mice. This was done by immunizing mice with inter-photoreceptor retinoid-binding protein (IRBP)$_{1-20}$ peptide (Fig. 1a). Compared with mice with mock induction, EAU-induced mice had

elevated gene expression of *Ghrh and Ghrhr* in the retina (Fig. 1b). Moreover, the gene expression of *Ghrhr splice variant 1* (*Sv1*), reported to express in normal human tissues and tumors[30,31], was also elevated in the retina of EAU mice (Fig. 1b). Consistently, immunofluorescence staining showed that GHRH and GHRH-R were markedly increased in ganglion cell layer and the inner nuclear layer of the retina (Fig. 1c). Collectively, the increased mRNA and protein of GHRH and GHRH-R in the retina of EAU mice suggested the GHRH-R signaling regulates responses to EAU induction. Alternatively, the enhanced expression of GHRH and GHRH-R in the retina could be a response to the inflammation.

### GHRH-R deficiency ameliorates autoimmune ocular inflammation
To investigate the impacts of GHRH and GHRH-R in uveitis, *Ghrhr^lit/lit* mice, with an amino acid substitution mutation D60G in the GHRH receptor resulting in the reduction of the GHRH-GH-IGF-1 signaling, were challenged with EAU[32,33]. As autoimmune uveitis is a chronic inflammatory disease in the eyes, we employed live ocular imaging technologies, confocal scanning laser ophthalmoscope (cSLO), and optical coherence tomography (OCT) to follow the morphological retinal changes in live mice[34]. cSLO and OCT examinations showed that GHRH-R deficient mice developed attenuated uveitis in terms of optic nerve head inflammation, vitreous and retinal infiltrates, retinal edema, and vasculitis (Fig. 1d–f and Supplementary Fig. 1a, b). More-over, quantified evaluation of retinal-choroidal thickness (RCT) by OCT and visions of photopic and scotopic B-wave amplitudes by electroretinography (ERG) confirmed that *Ghrhr^lit/lit* mice exhibited reduced fold change of RCT and better visions in dark and light adaption during the development of EAU, compared with wild-type (WT) mice (Supplementary Fig. 1a–d). Consistently, lower levels of histopathologic features in terms of chorioretinal infiltrates and extensive retinal structural damage corroborated with the alleviated uveitis in mice with deficient GHRH-R (Fig. 1g). We also immunized WT and *Ghrhr^lit/lit* mice with IRBP$_{651-670}$ peptide, which has been reported to induce more severe EAU[35]. Compared with WT mice, *Ghrhr^lit/lit* mice immunized with IRBP$_{651-670}$ developed less EAU in terms of disease score and quantified fold change of retinal-choroidal thickness (Sup-plementary Fig. 1e–h). In these *Ghrhr^lit/lit* mice, multiple cell types were deficient in GHRH-R. It is important to investigate whether lacking GHRH-R in the retina or in the effector cells contributed to the alle-viated uveitis in *Ghrhr^lit/lit* mice.

### GHRH-R deficiency impairs the pathogenicity of Th17 cells in autoimmune ocular inflammation
As EAU has been reported as a Th1 and Th17-mediated autoimmune disease[11], to evaluate the impact of GHRH-R deficiency during EAU, T cell subset in the spleen, eye-draining lymph nodes, and eyes were analyzed. Given the importance of the GHRH-R in the lymphopoiesis and thymic development[36,37], the effects of GHRH-R on the T cell development at the steady state (without EAU induction) was first investigated (Supplementary Fig. 2a–e). No alterations in the percen-tage of various T cell populations (CD4+, CD8+, CD4+CD8+, CD25+, AnnexinV+, CD44+CD62L-, IFN-γ+, IL-4+, IL-17A+, and Foxp3+) were found, indicating that GHRH-R exerted a minimal impact to the development and maintenance of T cells at the steady state, in agreement with previous studies[26]. After immunization, the activation of T cell was not affected (Supplementary Fig. 3a–d). However, there was a marked reduction in the percentage of CD4+ T cells expressing IL-17A, but not Th1, Th2, and Treg cell subsets, in spleen and eye-draining lymph nodes (Fig. 2a, b). The pathogenic IL-17A+IFN-γ+ double positive CD4+ T cells were also significantly lower in eye-draining lymph nodes and spleen (Fig. 2b and Supplementary Fig. 3i). No alterations in various T cell populations under the EAU-induced con-dition (CD4+, CD8+, CD25+, CD44+, and CD69+) were found

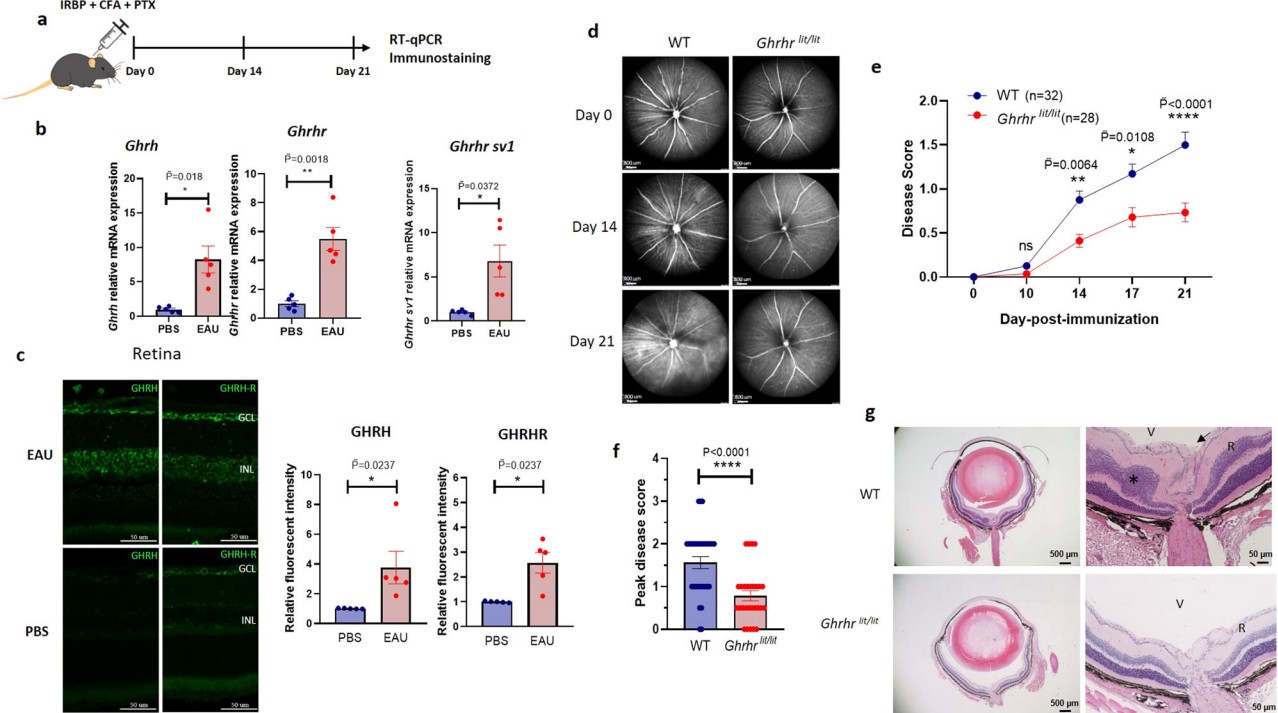

**Fig. 1 | Deficiency of GHRH-R protects mice from autoimmune ocular inflammation. a–c** WT mice were immunized with IRBP$_{1-20}$ peptide, complete Freund's adjuvant (CFA) and pertussis toxin (PTX) to induce EAU and were followed for 21 days. Retinae were collected at day 21 after immunization for quantitative real-time PCR (RT-qPCR) and immunostaining. **b** Relative gene expression of *Ghrh*, *Ghrhr*, and *Ghrhr Sv1* were quantified and normalized to *Gapdh*. Fold change is relative to control mice injected with PBS (*n* = 5). **c** GHRH (green color, left) and GHRH-R (green color, right) protein levels in the retina of EAU-induced mice and PBS controls were determined by immunostaining and quantified by ImageJ (*n* = 5). GCL ganglion cell layer, INL inner nuclear layer. Scale bar: 50 µm. **d–g** WT mice (*n* = 32) and *Ghrhr^{lit/lit}* mice (*n* = 28) were challenged with EAU and assessed at five time points throughout 21 days. **d** Representative fundus images by cSLO at three time points after immunization. Scale bar: 800 µm. **e** Disease scores of EAU were graded throughout 21 days. The exact *p* value on Day 21 was $1.97 \times 10^{-6}$. **f** Peak disease scores of uveitis throughout 21 days. The exact *p* value was $9.81 \times 10^{-5}$. **g** At day 21, mice were sacrificed and eyeball were collected for H&E staining. Eye-infiltrating cells (arrow) and retinal fold (asterisk) are indicated (*n* = 5). V vitreous, R retina. Scale bars: 500 µm (left) and 50 µm (right). Data were the representation of at least two independent experiments. Data were presented as mean ± SEM. Student's *t*-test (**f**) and Mann–Whitney test with Bonferroni correction (**b**, **c**, **e**). Statistical tests were all two-sided (**b**, **c**, **e**, **f**). *, ** and **** represent $\widetilde{P} < 0.05$ or $\widetilde{P} < 0.01$ and $\widetilde{P}$ (or *P*) < 0.0001 respectively. ns represents no significant difference.

(Supplementary Fig. 3a–d). To further investigate the pathogenicity of GHRH-R deficient Th17 cells in EAU, CD4$^+$ T cells isolated from the uveitic eye were analyzed. GHRH-R deficient mice showed a significantly lower frequency of IL-17A$^+$CD4$^+$ T cells co-expressing the pathogenic Th17 cytokines GM-CSF or IFN-γ in their eyes (Fig. 2c). Accordingly, mRNA levels in genes associated with pathogenic Th17 cells, including *Il17a, Il17f, Il22*, and *Csf2* were decreased, while *Il10* was increased, in the CD4$^+$ T cells isolated from the eyes of *Ghrhr^{lit/lit}* mice. There is no difference in the expression of gene C-C motif chemokine receptor 6 (*Ccr6*), a chemokine receptor critical for Th17 cell migration into the inflamed tissues[38] (Fig. 2d), indicating GHRH-R deficiency did not alter the migration of these eye-infiltrating CD4$^+$ T cells.

To further confirm the impact of GHRH-R on the pathogenicity of IRBP-specific Th17 cells in uveitis, the capability of pathogenic *Ghrhr^{lit/lit}* Th17 cells to induce uveitis was assessed by adoptive transfer into recipient mice without IRBP injection. Donor WT and *Ghrhr^{lit/lit}* mice were immunized with IRBP$_{1-20}$ and after 10 days, CD4$^+$ T cells were purified from the spleen and draining lymph nodes and cultured under Th17 polarizing condition with IRBP$_{1-20}$ for 3 days. Then the equal amount of live CD4$^+$ T cells were transferred into naïve WT or *Ghrhr^{lit/lit}* recipient mice (Fig. 2e). Both WT and *Ghrhr^{lit/lit}* recipient mice receiving the IRBP-specific T cells from WT donor mice were able to develop a similar onset and severity of uveitis (Fig. 2f and Supplementary Fig. 3e–h). However, the IRBP-specific T cells isolated from *Ghrhr^{lit/lit}* donor mice could not elicit uveitis in WT or *Ghrhr^{lit/lit}* recipient mice (Fig. 2f and Supplementary Fig. 3e–h). These data indicate GHRH-R

deficiency impaired the pathogenicity of Th17 cells in autoimmune uveitis, despite wild-type GHRH-R was still expressed in the retina.

## Th17 cells express GHRH-R throughout Th17 differentiation

To investigate the impacts of GHRH-R in T cells, mRNA expression of *Ghrhr* and *Sv1* was initially measured in T cell subsets. Naïve T cells were isolated and gated as CD4$^+$CD25$^-$CD62L$^+$CD44$^-$ T cells (Supplementary Fig. 4a), followed by in vitro differentiation into Th0, Th1, Th2, Th17, and induced regulatory T (iTreg) cells. Gene expression of *Ghrhr and Sv1* was constitutively expressed at low level in naïve T cells and was upregulated in all T cell subsets at 48 h under selective polarizing conditions (Fig. 3a and Supplementary Fig. 4d). Notably, Th17 cells showed a significant increase in mRNA expression of *Ghrhr* and *Sv1* compared with Th0 cells (Fig. 3a and Supplementary Fig. 4d). Furthermore, during Th17 cell differentiation along various time points, *Ghrhr* gene expression was increasingly upregulated from 12 h, peaking at 48 h, and then declined at 96 h in comparison to Th0 cells, which showed a constantly lower level of *Ghrhr* along 96 h (Fig. 3b). *Ghrhr* gene expression in cells under Th1, Th2, or iTreg conditions showed a slight increase from 12 h and peaked at 48 h, followed by a gradual decrease to the original levels at 96 h (Supplementary Fig. 4e). Immunofluorescence staining confirmed that GHRH-R was hardly detected in naïve T cells, whereas it was expressed and localized predominantly on the cell membrane of Th17 cells (Fig. 3c). In addition, flow cytometry analysis showed the GHRH-R expression in the IL-17A$^+$CD4$^+$ T cells was higher than that in the IL-17A$^-$CD4$^+$ T cells from

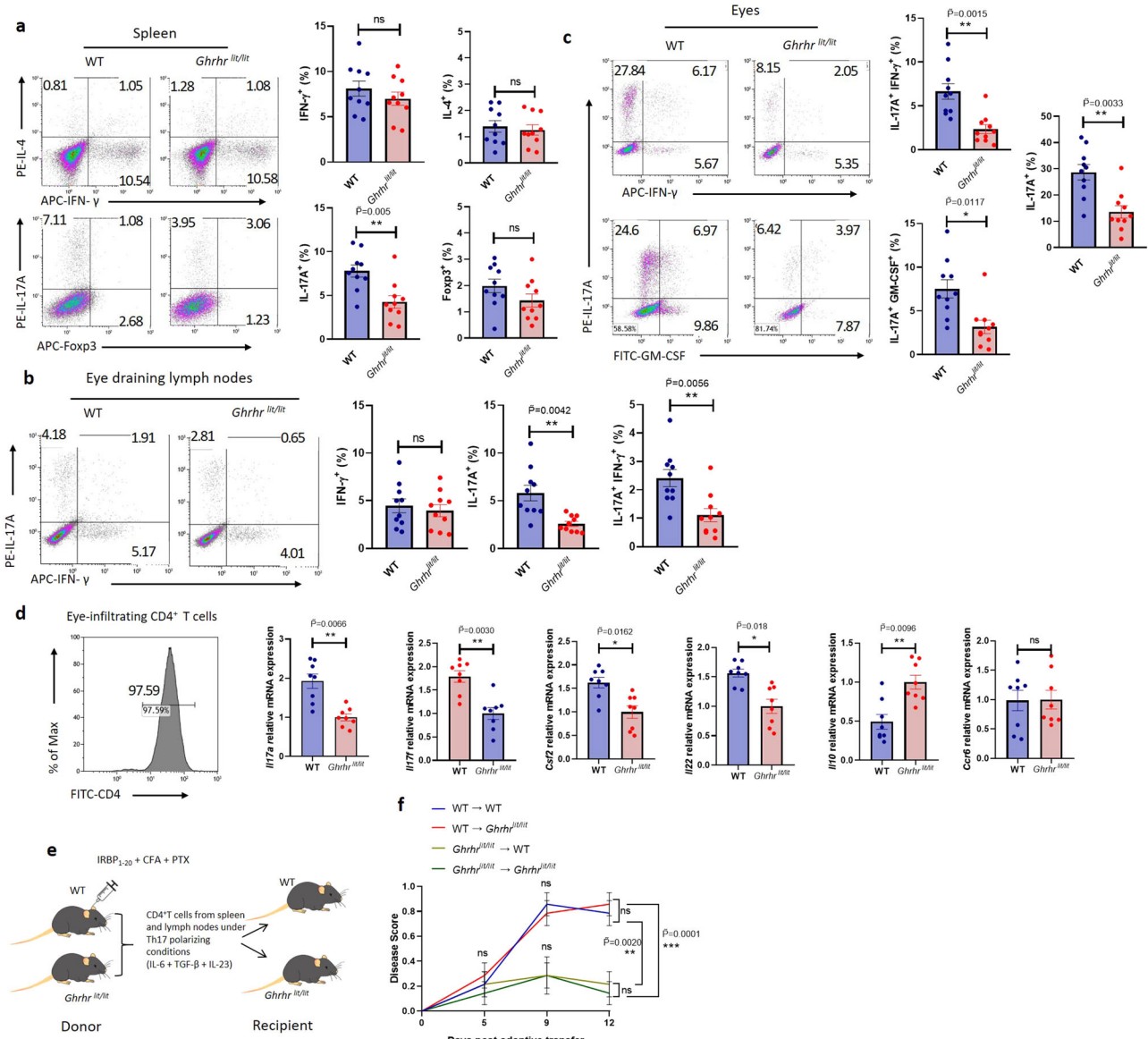

**Fig. 2 | Deficiency of GHRH-R protects mice from autoimmune ocular inflammation by inhibiting the pathogenicity of Th17 cells.** WT and *Ghrhr^lit/lit* mice were challenged with EAU. After 21 days, cells were isolated from the spleen, eye-draining lymph nodes, and eyes for flow cytometry analysis (*n* = 10). **a** IFN-γ, IL-4, IL-17A, and Foxp3 expression in CD4+ T cells from the spleen. **b** IFN-γ and IL-17A expression in CD4+ T cells from eye-draining lymph nodes. **c** IL-17A+CD4+ T cells from eyes co-expressing GM-CSF or IFN-γ. **d** Eye-infiltrating CD4+ T cells were enriched and verified by flow cytometry, followed by the analysis of gene expression of *Il17a*, *Il17f*, *Csf2*, *Il22*, *Il10*, and *Ccr6* (4–6 eyeballs were pooled as one sample, *n* = 8). **e** CD4+ T cells from IRBP-immunized donor WT and *Ghrhr^lit/lit* mice were cultured

under Th17 cell-polarizing conditions (IL-6, TGF-β, and IL-23) with IRBP$_{1-20}$ peptide for 3 days and adoptively transferred into naïve WT and *Ghrhr^lit/lit* recipient mice. **f** Disease scores were graded at four time points post-transfer (*n* = 7). The *F* value of the one-way ANOVA test is 14.91 on day 12, and the corresponding *p* value is less than 0.0001. Data were the representation of at least two independent experiments. Data were presented as mean ± SEM. Student's *t*-test with Bonferroni correction (**a**–**d**) and one-way ANOVA followed by Bonferroni post hoc test (**f**). Statistical tests were all two-sided (**a**–**d**). *, **, and *** represent $\tilde{P}$ < 0.05, $\tilde{P}$ < 0.01, and $\tilde{P}$ = 0.001, respectively. ns represents no significant difference.

---

the same samples (Fig. 3d). Taken together, these data indicated GHRH-R, similar to IL-23R, is rarely expressed in naïve T cells, while it is expressed in Th17 cells during differentiation[39]. However, unlike IL-23R being specifically expressed in Th17 cells, GHRH-R is also expressed in other T cell subsets, indicating that the expression of GHRH-R during differentiation of all T cells is not only induced by the signaling factors (IL-6 and TGF-β) required for the differentiation into Th17 cells.

**GHRH-R is critical to Th17 differentiation**
To investigate whether GHRH-R is critical to T cell differentiation, naïve CD4+ T cells from WT and *Ghrhr^lit/lit* mice were activated with anti-CD3 and anti-CD28 antibodies and cultured under Th1, Th2, Th17, or

iTreg cell-polarizing conditions. GHRH-R-deficient naïve T cells had a reduced capacity for differentiation into Th17 cells, as evidenced by lower expression of IL-17A and RORγt (Fig. 4a–d). Meanwhile, GHRH-R deficiency did not affect T cell differentiation into Th1, Th2, or iTreg cells, which was assessed by the expression of the cytokines IFN-γ and IL-4, and transcription factors T-bet, GATA3, and Foxp3 (Fig. 4a–d).

Th17 cells generated in vitro in stimulation with TGF-β and IL-6 were reported to be non-pathogenic due to their incapability to provoke autoimmune disease after transferring into mice[10]. However, additional in vitro exposure to IL-23 is able to render these cells pathogenic, as does their differentiation in the absence of TGF-β, with a combination of IL-6, IL-1β, and IL-23[9,10]. Given GHRH-R deficiency

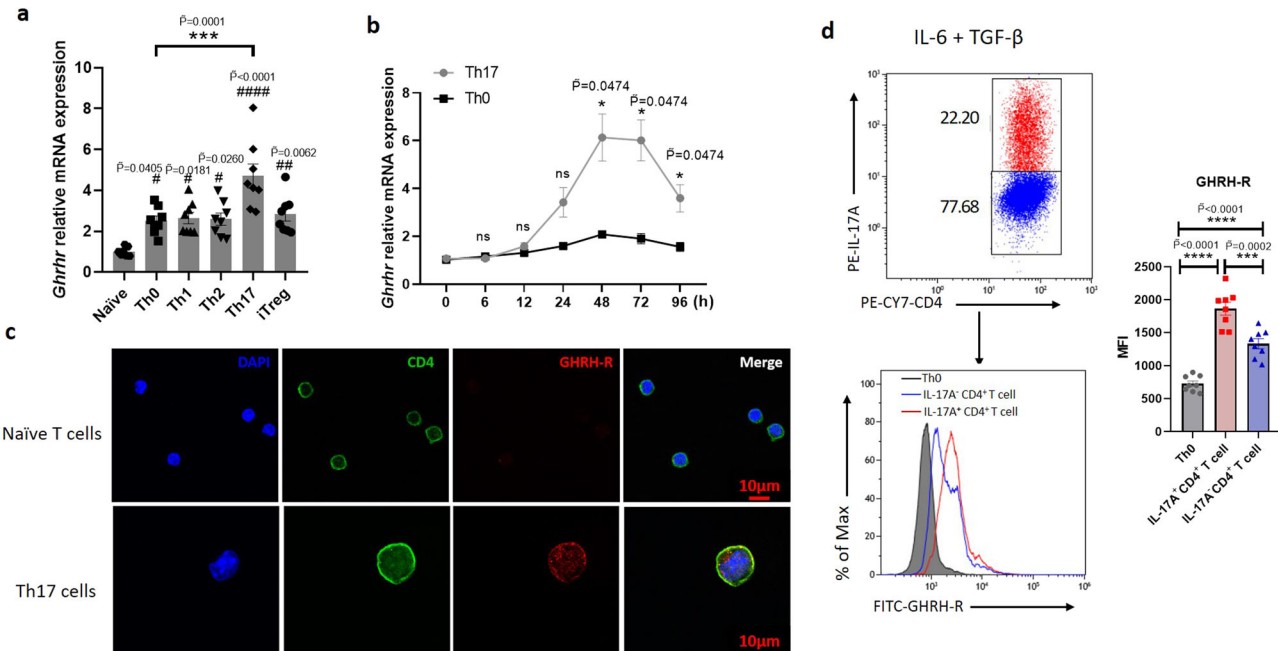

**Fig. 3 | GHRH-R expressed in Th17 cells throughout differentiation. a** Naïve CD4+ T cells were cultured in the stimulation with anti-CD3 and anti-CD28 antibodies in T cell differentiation condition: None (Th0), IL-12 (Th1), IL-4 (Th2), IL-6 and TGF-β (Th17), TGF-β (iTreg). After 48 h, *Ghrhr* gene expression was quantified and normalized to *Gapdh*; fold change is relative to naïve controls ($n = 8$). The *F* value of the one-way ANOVA test is 12.34, and the corresponding *p* value is less than 0.0001. The exact *p* value (naïve vs Th17) was $1.64 \times 10^{-8}$. **b** Naïve CD4+ T cells were activated (Th0) and differentiated into Th17 cells. *Ghrhr* gene expression was evaluated at seven time points ($n = 5$). **c** Staining of GHRH-R (red) and CD4 (green) was performed in differentiated Th17 cells; nuclei were labeled with DAPI (blue). Scale bar: 10 μm. **d** Naïve T cells were differentiated into Th17 cells for flow cytometry analysis of GHRH-R expression in the CD4+IL-17A+ and CD4+IL-17A- T cells ($n = 8$). MFI, mean fluorescence intensity. The *F* value of the one-way ANOVA test is 56.39, and the corresponding *p* value is less than 0.0001. The exact *p* values (Th0 vs Th17+CD4+ and Th0 vs Th17+CD4-) were $2.03 \times 10^{-9}$ and $3.82 \times 10^{-5}$, respectively. Data were the representation of at least two independent experiments. Data were presented as mean ± SEM. Two-sided Mann–Whitney test with Bonferroni correction (**b**) and one-way ANOVA followed by Bonferroni post hoc test (**a**, **d**). #, ##, and #### represent $\widetilde{P} < 0.05$, $\widetilde{P} < 0.01$, and $\widetilde{P} < 0.0001$ when compared to naïve T cells. *, ***, and **** represent $\widetilde{P} < 0.05$, $\widetilde{P} < 0.001$, and $\widetilde{P} < 0.0001$, respectively. ns represents no significant difference.

impaired the differentiation of non-pathogenic Th17 cells, we further examined its impact on the differentiation of Th17 under two different pathogenic conditions: (1) IL-6, TGF-β, and IL-23; (2) IL-6, IL-1β, and IL-23. Our results showed that *Ghrhr*^*lit/lit*^ T cells had reduced IL-17A production after differentiating in both pathogenic Th17 conditions, compared with WT. (Fig. 4e, f). To further address the mechanism whereby GHRH-R deficiency impaired Th17 cell differentiation without any additional treatment of GHRH ligands in the cell culture medium, we investigated whether GHRH, the natural ligand for GHRH-R, was produced by Th17 cells to enhance their own differentiation. Unexpectedly, the expression of *Ghrh* was not detectable from Th17 cells under the stimulation of IL-6 with TGF-β or IL-6, IL-1β with IL-23. Additionally, other cytokine receptors associated with Th17 cell differentiation were also measured. *Gp130*, *Il1r*, and *Il21r* expression showed no significant changes in *Ghrhr*^*lit/lit*^ Th17 cells, while *Il23r* expression was decreased in *Ghrhr*^*lit/lit*^ T cells under stimulation with IL-6 and TGF-β or IL-6, IL-1β, and IL-23 (Fig. 4g, h). Moreover, GH and IGF-1 have been reported to regulate T cell development[40,41]. To investigate the influence on T cell differentiation by GHRH-R downstream molecules GH and IGF-1 produced by the T cell itself, antibodies against GHRH, GH, and IGF-1 were applied to neutralize these molecules. Significantly decreased expression of IL-17A were observed in WT Th17 cells, but not in *Ghrhr*^*lit/lit*^ Th17 cells, in the presence of anti-GHRH antibody during differentiation, indicating that GHRH in the cell culture environment could influence the differentiation of Th17 cells (Fig. 5a). However, neutralization by antibodies against GH and IGF-1 did not alter IL-17A production between WT and *Ghrhr*^*lit/lit*^ Th17 cells (Fig. 5a).

We next examined the influences of GHRH-R signaling in the naïve CD4+ T cell differentiation into Th17 cells with or without exogenous GHRH-R ligands by using human GHRH (hGHRH), GHRH agonist MR-409 and antagonist MIA-602. Both MR-409 and MIA-602 exhibit much higher affinity in receptor binding and a longer half-life in serum compared with hGHRH[42,43]. Naïve CD4+ T cells isolated from WT and *Ghrhr*^*lit/lit*^ mice were cultured with hGHRH, MR-409, and MIA-602 under Th17 cell-polarizing conditions. hGHRH significantly enhanced the differentiation into Th17 cells in WT cells upon the stimulation with IL-6 and TGF-β, as did GHRH agonist, which showed a more potent effect. Nevertheless, *Ghrhr*^*lit/lit*^ Th17 cells showed no response to hGHRH or GHRH agonist over differentiation. GHRH antagonist reduced Th17 cell differentiation in WT cells to a comparable level in *Ghrhr*^*lit/lit*^ cells (Fig. 5b). Moreover, these GHRH analogs exhibited dose-dependent effects (Supplementary Fig. 5a–c). Consistently, the Th17-specific effect was observed in both GHRH agonist and antagonist as both of which failed to influence the Th1, Th2, and iTreg cell differentiation (Supplementary Fig. 5d, e). Collectively, these data indicate that the signaling from GHRH and its agonist binding to GHRH-R promotes the differentiation into Th17 cells, which can be abolished by blocking its signaling using an anti-GHRH antibody or GHRH antagonist. Only low expression levels of IFN-γ could be detected under these Th17 cell-polarizing conditions. Therefore pathogenic Th17 cells could not be quantified by co-staining cells with IL-17A and IFN-γ (Supplementary Fig. 5l–m), which was different from the in vivo data. Furthermore, we were unable to detect the expression of IL-9, a regulator of pathogenicity in Th17 cells[44], whereas IL-10 expression is significantly higher in GHRH-R deficient cells than in WT cells

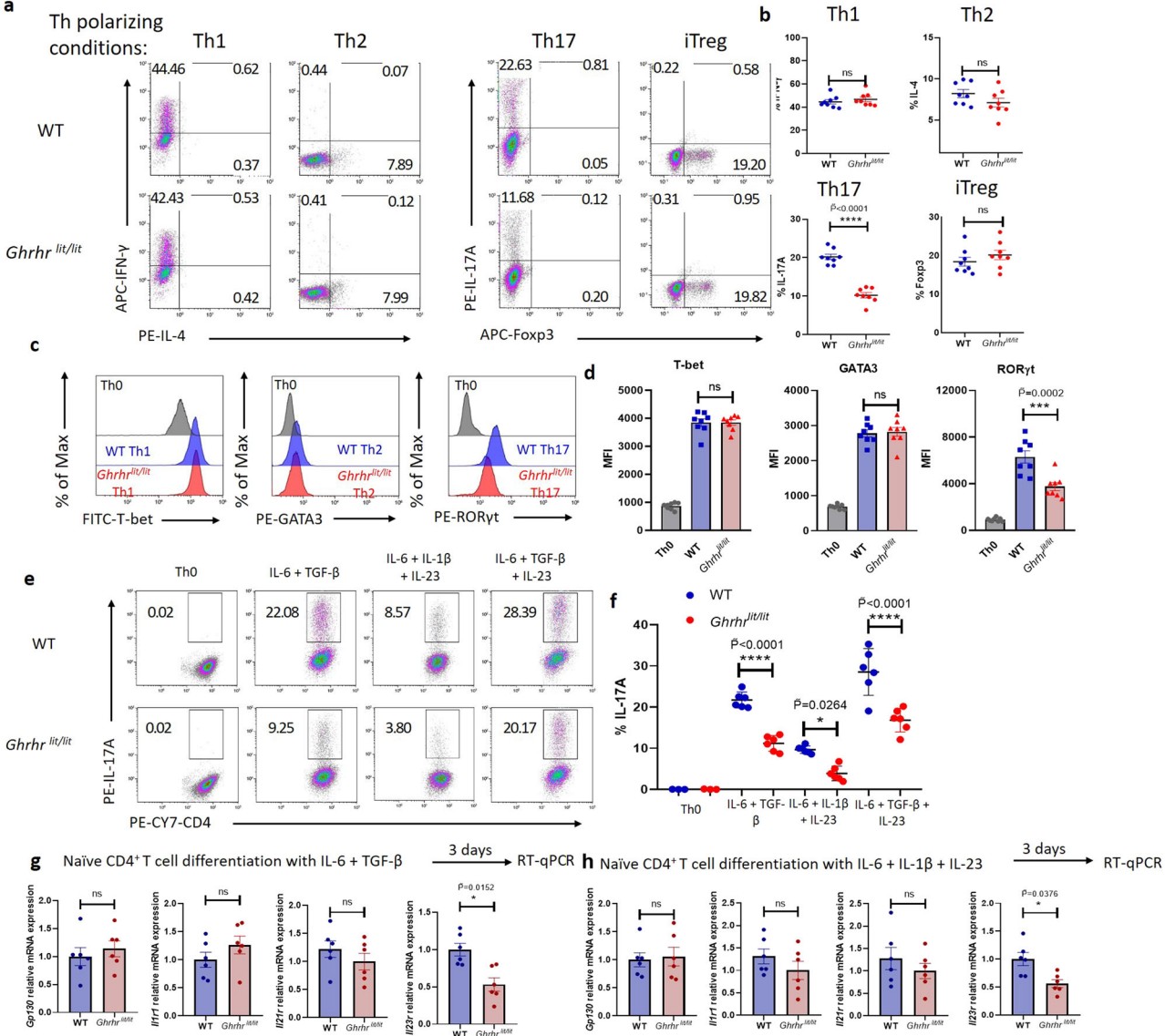

**Fig. 4 | GHRH-R is required for Th17 cell differentiation in vitro. a–d** Naïve CD4+ T cells from WT and *Ghrhr^lit/lit^* mice were cultured in Th0, Th1, Th2, Th17, or iTreg polarizing conditions for 3 days (*n* = 8) and evaluated by flow cytometry for the expression of cytokines IFN-γ, IL-4, and IL-17A for Th1, Th2, and Th17, respectively; and transcription factor T-bet, GATA3, RORγt and Foxp3 for Th1, Th2, Th17, and iTreg respectively. MFI mean fluorescence intensity. The exact *p* value (Th17) was 2.09 × 10⁻⁸. **d** The *F* value of the one-way ANOVA test is 55.46, and the corresponding *p* value is less than 0.0001. **e** Naïve CD4+ T cells from WT and *Ghrhr^lit/lit^* mice were cultured for 3 days in non-pathogenic Th17 polarizing condition (IL-6 and TGF-β), or pathogenic Th17 polarizing condition: (1) IL-6, TGF-β, and IL-23; (2) IL-6, IL-1β, and IL-23. **f** Frequency of IL-17A+CD4+ T cells was determined by flow cytometry (*n* = 6). The *F* value of the one-way ANOVA test is 67.21, and the

corresponding *p* value is less than 0.0001. The exact *p* values (IL-6 + TGFβ WT vs *Ghrhr^lit/lit^* and IL-6 + TGFβ + IL23 WT vs *Ghrhr^lit/lit^*) were 2.03 × 10⁻¹¹ and 4.54 × 10⁻⁷ respectively. **g, h** Naïve CD4+ T cells from WT and *Ghrhr^lit/lit^* were cultured in non-pathogenic (IL-6 and TGF-β) or pathogenic Th17 cell differentiation condition (IL-6, IL-1β, and IL-23) for 3 days. Expression of *Gp130, Il1r1, Il21r*, and *Il23r* was quantified and normalized to *Gapdh* (*n* = 6). Fold change is relative to the *Ghrhr^lit/lit^* group. Data were the representation of at least two independent experiments. Data were presented as mean ± SEM. Two-sided student's *t*-test with Bonferroni correction (**b, g, h**) and one-way ANOVA followed by Bonferroni post hoc test (**d, f**). *, ***, and **** represent P̃ < 0.05, P̃ < 0.001, and P̃ < 0.0001 respectively. ns represents no significant difference.

(Supplementary Fig. 5n). To further investigate the effects of GHRH-R signaling in Th17 cells, naïve T cell from WT and *Ghrhr^lit/lit^* mice were polarized into Th17 cells for 3 days in vitro. The resulting cells were starved in a serum-free medium for 4 h and then exposed to GHRH agonist for 30–60 min, followed by RT-qPCR analysis. The expression of genes associated with pathogenic Th17 cells, including *Il17a, Il17f, Il22, Il23r, Rorc, Rora,* and *Hif1a* were reduced in *Ghrhr^lit/lit^* Th17 cells, whereas *Batf* and *Irf4* were not altered (Fig. 5c). We further verified our results in the *Ghrhr* knocked down cells. siRNA knocked down of *Ghrhr* in WT naïve T cells led to less differentiation towards Th17 cells (Supplementary Fig. 5k). These data suggest that GHRH-R signaling

promoted, while inhibition of GHRH-R signaling reduced, the differentiation and pathogenicity of Th17 cells.

To assess whether GHRH-R deficiency affects T cell division, proliferation, viability, and activation in vitro, we activated naïve CD4+ T cells from WT and *Ghrhr^lit/lit^* mice with anti-CD3 and anti-CD28 under Th17 polarizing conditions with or without the addition of GHRH agonist for 3–5 days. First, we analyzed the dilution of CellTrace violet dye after Th17 cell culture for 3 days. The vast majority of WT and *Ghrhr^lit/lit^* Th17 cells with or without additional GHRH agonist entered the cell division (Fig. 5d). Additionally, WT and *Ghrhr^lit/lit^* Th17 cells, with or without additional GHRH agonist, no significant difference in

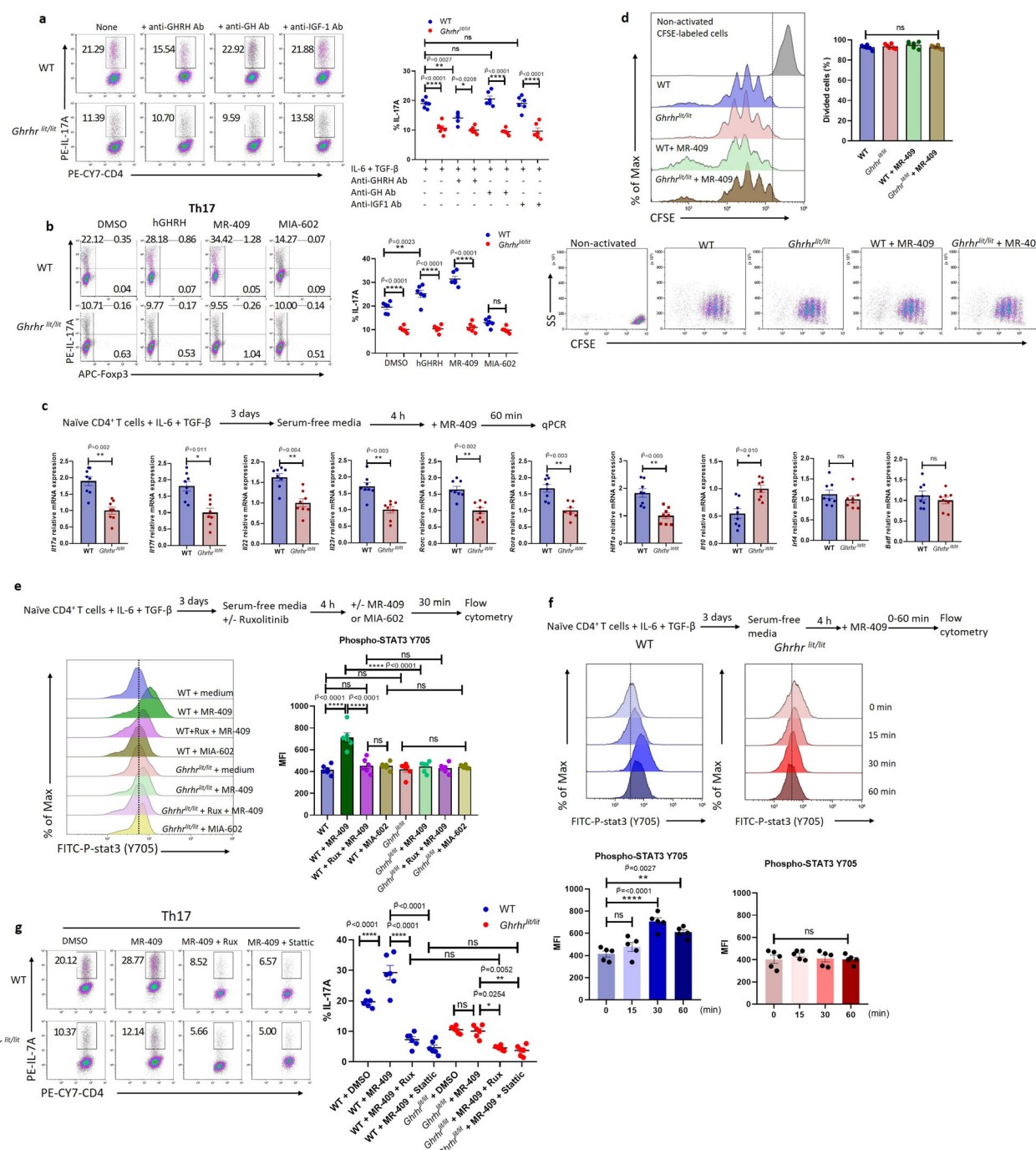

cell viability (Supplementary Fig. 5f, g) and activation mediated by T cell receptor (TCR) were observed, as indicated by a similar expression of markers of TCR activation (CD44, CD25, and CD69) (Supplementary Fig. 5h–j). Our results suggested GHRH-R deficiency does not affect T cell division, proliferation, viability, and activation.

## GHRH signaling promotes STAT3 phosphorylation in Th17 cell differentiation

The aberrant Janus kinase (JAK)-STAT3 signaling has been reported in various immune disorders[5,45]. Our previous study demonstrated that GHRH-R signaling through the JAK2-STAT3 pathway directly mediated the inflammatory responses to LPS insults in the human ciliary body and mouse iris[21]. To investigate whether GHRH-R signaling mediates Th17 cell differentiation via the JAK-STAT3 pathway, naïve T cells from WT and *Ghrhr^lit/lit^* mice were cultured in Th17 polarizing conditions for

3 days. Then these cells were starved in the serum-free medium with or without ruxolitinib (Rux), a JAK 1/2 inhibitor, for 4 h prior to a pulse with GHRH agonist or antagonist for 30 min. The phosphorylation of STAT3 (p-STAT3) Y705 was determined by flow cytometry. The results showed no significant difference of p-STAT3 Y705 between WT and *Ghrhr^lit/lit^* Th17 cells without stimulation by GHRH ligands (Fig. 5e). After a pulse with GHRH agonist, WT Th17 cells had increased p-STAT3 Y705, which was decreased by blocking JAK1/2 with Rux, compared with *Ghrhr^lit/lit^* Th17 cells. As expected, the GHRH antagonist failed to elevate p-STAT3 Y705 in both WT and *Ghrhr^lit/lit^* Th17 cells (Fig. 5e). We next sought to investigate the time-dependent effects of GHRH signaling in the p-STAT3 Y705 in Th17 cells from 0–60 min. The p-STAT3 Y705 in WT Th17 cells had been steadily increasing from 15 min after the pulse with GHRH agonist, peaked at 30 min, and followed by a slight decrease at 60 min. Conversely, *Ghrhr^lit/lit^* Th17 cells had no

**Fig. 5 | GHRH-R signaling promotes Th17 cell differentiation via the JAK-STAT3 pathway. a, b** Naïve CD4$^+$ T cells from WT and *Ghrhr$^{lit/lit}$* mice were cultured in Th17 polarizing condition for 3 days. **a** 10 µg/ml of anti-GHRH, anti-GH or anti-IGF-1 antibodies were added to cells for 3 days. The frequency of IL-17A$^+$CD4$^+$ T cells was determined by flow cytometry (*n* = 6 independent experiments). The *F* value of one-way ANOVA test is 36.15, and the corresponding *p* value is less than 0.0001. The exact *p* values (IL-6 + TGFβ WT vs *Ghrhr$^{lit/lit}$*, anti-GH WT vs *Ghrhr$^{lit/lit}$*, and anti-IGF-1 WT vs *Ghrhr$^{lit/lit}$*) were $1.43 \times 10^{-7}$, $8.24 \times 10^{-11}$, and $7.07 \times 10^{-9}$ respectively. **b** DMSO, hGHRH, MR-409, or MIA-602 were added to cells for 3 days. The frequency of IL-17A$^+$CD4$^+$ T cells was determined by flow cytometry (*n* = 6 independent experiments). The *F* value of the one-way ANOVA test is 79.74, and the corresponding *p* value is less than 0.0001. The exact *p* values (DMSO WT vs *Ghrhr$^{lit/lit}$*, hGHRH WT vs *Ghrhr$^{lit/lit}$*, and MR-409 WT vs *Ghrhr$^{lit/lit}$*) were $2.35 \times 10^{-7}$, $1.63 \times 10^{-12}$, and $4.71 \times 10^{-13}$, respectively. **c** Naïve CD4$^+$ T cells from WT and *Ghrhr$^{lit/lit}$* mice were activated for 3 days in Th17 polarizing condition, washed, and serum-starved for 4 h and then re-stimulated with MR-409 for 60 min. Expression of *Il17a, Il17f, Il22, Il23r, Rorc, Rora, Hif1a, Il10, Irf4,* and *Batf* was quantified (*n* = 8 independent experiments). Data were normalized to *Gapdh* and fold change is relative to *Ghrhr$^{lit/lit}$*. **d** Naïve CD4$^+$ T cells from WT and *Ghrhr$^{lit/lit}$* mice were labeled with CellTrace dye (CFSE). Cells were cultured in Th17 polarizing condition with or without MR-409 for 3 days (*n* = 6

independent experiments). **e** Then cells were treated with or without ruxolitinib (1 µM) for 4 h. MR-409 and MIA-602 were then added to cells for 30 min. Expression of phosphorylation of STAT3 Y705 in CD4$^+$IL-17$^+$ T cells was quantified (*n* = 6 independent experiments). The *F* value of the one-way ANOVA test is 17.77, and the corresponding *p* value is less than 0.0001. **f** MR-409 was treated to cells for 0, 15, 30, and 60 min. Expression of phosphorylation of STAT3 Y705 in CD4$^+$IL-17$^+$ T cells was quantified (*n* = 6 independent experiments). The *F* value of the one-way ANOVA test is 17.38, and the corresponding *p* value is less than 0.0001. **g** Naïve CD4$^+$ T cells from WT and *Ghrhr$^{lit/lit}$* mice were cultured for 3 days in Th17 polarizing condition in the presence of MR-409 with or without ruxolitinib (1 µM) or Stattic (1 µM) for 3 days. The frequency of IL-17A$^+$CD4$^+$ T cells was determined by flow cytometry (*n* = 6 independent experiments). The *F* value of the one-way ANOVA test is 67.11, and the corresponding *p* value is less than 0.0001. The exact *p* values (WT + DMSO vs WT + MR-409, WT + MR-409 vs WT + MR-409+Rux, and WT + MR-409 vs WT + MR-409+Stattic) were $7.62 \times 10^{-6}$, $1 \times 10^{-15}$, and $1 \times 10^{-15}$ respectively. Data were the representation of at least two independent experiments. Data were presented as mean ± SEM. Two-sided student's *t*-test with Bonferroni correction (**c**) and one-way ANOVA followed by Bonferroni post hoc test (**a, b, e–g**). *, **, and **** represent $\tilde{P} < 0.05$, $\tilde{P} < 0.01$, and $\tilde{P} < 0.0001$, respectively. ns represents no significant difference.

response to the pulse with GHRH agonist (Fig. 5f). Finally, we determined the importance of the GHRH-R-JAK-STAT3 pathway mediating Th17 cell differentiation. Blocking of JAK1/2 by Rux or inhibiting the activation of STAT3 by a Stattic completely abolish GHRH agonist-induced signaling, resulting in significantly reduced production of IL-17A in WT CD4$^+$ T cell during differentiation (Fig. 5g). These data demonstrated that GHRH-R signaling influences Th17 cell differentiation via GHRH-R-JAK-STAT3 pathway.

## GHRH agonist promotes, while GHRH antagonist ameliorates, Th17-mediated ocular inflammation

Our results showed that GHRH-R is required for Th17 cell differentiation and Th17 cell-mediated autoimmune ocular inflammation via the use of genetically modified mice with mutated GHRH-R. To further confirm these findings from a pharmacological perspective, GHRH agonist MR-409 and antagonist MIA-602 were injected daily from day 10 to day 21 after immunization into WT mice with EAU induction. Clinical examinations by cSLO and OCT, and ocular histology showed that compared with the vehicle group (EAU + DMSO), mice treated with GHRH agonist (EAU + MR-409) developed more severe uveitis, while mice treated with GHRH antagonist (EAU + MIA-602) produced less uveitis (Fig. 6a, b and Supplementary Fig. 6a–c). Consistently, qRT-PCR analysis of RNA from isolated retina revealed that compared with the vehicle group (EAU + DMSO), the expression of cytokines for pathogenic Th17 cells, including *Il17a, Il17f,* and *Cfs2* were upregulated by GHRH agonist, while downregulated by GHRH antagonist (Fig. 6d). Although the expression of *Ghrh* was elevated significantly in the retina of mice immunized with IRBP compared with mice mock treated with PBS, neither GHRH agonist nor antagonist altered the expression of *Ghrh* in these retinae (Fig. 6d). To further confirm whether GHRH agonist or antagonist mediated the autoimmune ocular inflammation by regulating pathogenic Th17 cells in vivo, we examined the population of T cell subsets, as well as T cell activation and apoptosis in the spleen, draining lymph nodes, and eyes. Consistently, no alteration was observed in the population of CD4$^+$ and CD8$^+$ T cells and the markers of T cell activation, including CD25, CD69, CD44, and CD62L (Supplementary Fig. 6d–g). Consistent with our in vitro experimental results, IRPB$_{1-20}$ immunized mice treated with GHRH agonist have significantly increased IL-17A$^+$CD4$^+$ T cells but not other T cell subsets in the spleen, eye-draining lymph nodes, and eyes. Conversely, GHRH antagonist-treated mice have decreased IL-17A$^+$CD4$^+$ T cells, as well as the pathogenic Th17 cells (Fig. 6e–g and Supplementary Fig. 6h). In addition, in the eye, the pathogenic Th17 cells expressing IL-17A$^+$GM-CSF$^+$ or IL-17A$^+$IFN-γ$^+$ were also increased in the GHRH agonist treated

mice but were reduced in the GHRH antagonist-treated mice (Fig. 6g). We next enriched the CD4$^+$ T cells from the eyes and analyzed the expression of genes associated with pathogenic Th17 cells. The expression of pathogenic cytokines of Th17 cells, including *Il17a, Il17f, Il22,* and *Csf2,* were upregulated in mice treated with GHRH agonist and were downregulated by GHRH antagonist (Fig. 6h). The expression of *Ccr6* was not altered by GHRH agonist and antagonist (Fig. 6h). Consistent with data from GHRH-R deficient mice challenged with uveitis, GHRH antagonist protected mice from uveitis as well, while GHRH agonist promoted it. Collectively, our data confirmed that GHRH signaling is required for the differentiation of Th17 cells in autoimmune ocular inflammation.

## GHRH-R deficiency ameliorates autoimmune neuroinflammation

Given the importance of GHRH signaling in Th17 differentiation and Th17 cell-mediated autoimmune ocular inflammation, we next addressed if this pathway also mediates other Th17 cell-driven autoimmune diseases. WT and *Ghrhr$^{lit/lit}$* mice were immunized with myelin oligodendrocyte glycoprotein (MOG)$_{35-55}$ peptide to induce EAE, a mouse model mimicking human multiple sclerosis[46]. Mice with deficient GHRH-R produced reduced inflammation in the central nervous system (CNS) in terms of body weight loss, peak disease score, and histopathology (Supplementary Fig. 7a–d). Importantly, there was a significant reduction in CD4$^+$ IL17A$^+$ T cells in the draining lymph nodes, spinal cord, and brain in *Ghrhr$^{lit/lit}$* EAE mice (Supplementary Fig. 7e–g). In addition, the CNS-pathogenic Th17 cells (IL-17$^+$IFN-γ$^+$ or IL-17A$^+$GM-CSF$^+$) were decreased significantly in the CNS (Supplementary Fig. 7f, g). Altogether, these data indicated that GHRH-R deficiency protects mice from Th17-mediated autoimmune neuroinflammation.

## Discussion

The neuroendocrine system and the immune system are known to communicate with each other. Ligands and their receptors have been reported to be shared by cells of both systems[47]. Recently, the receptor of GHRH, a neuropeptide secreted from the hypothalamus, has been found on T cells, and upon binding to its ligands, the synthesis and secretion of GH from T cells are induced, which can serve as a para-crine/autocrine regulation to indirectly facilitate immune responses[48]. However, the impacts of GHRH-R in T cells remain not very well understood. In this study, *Ghrhr$^{lit/lit}$* mice were employed to study the influence of GHRH-R signaling in Th17 cell differentiation genetically. In addition, to eliminate the underlying alterations in immune

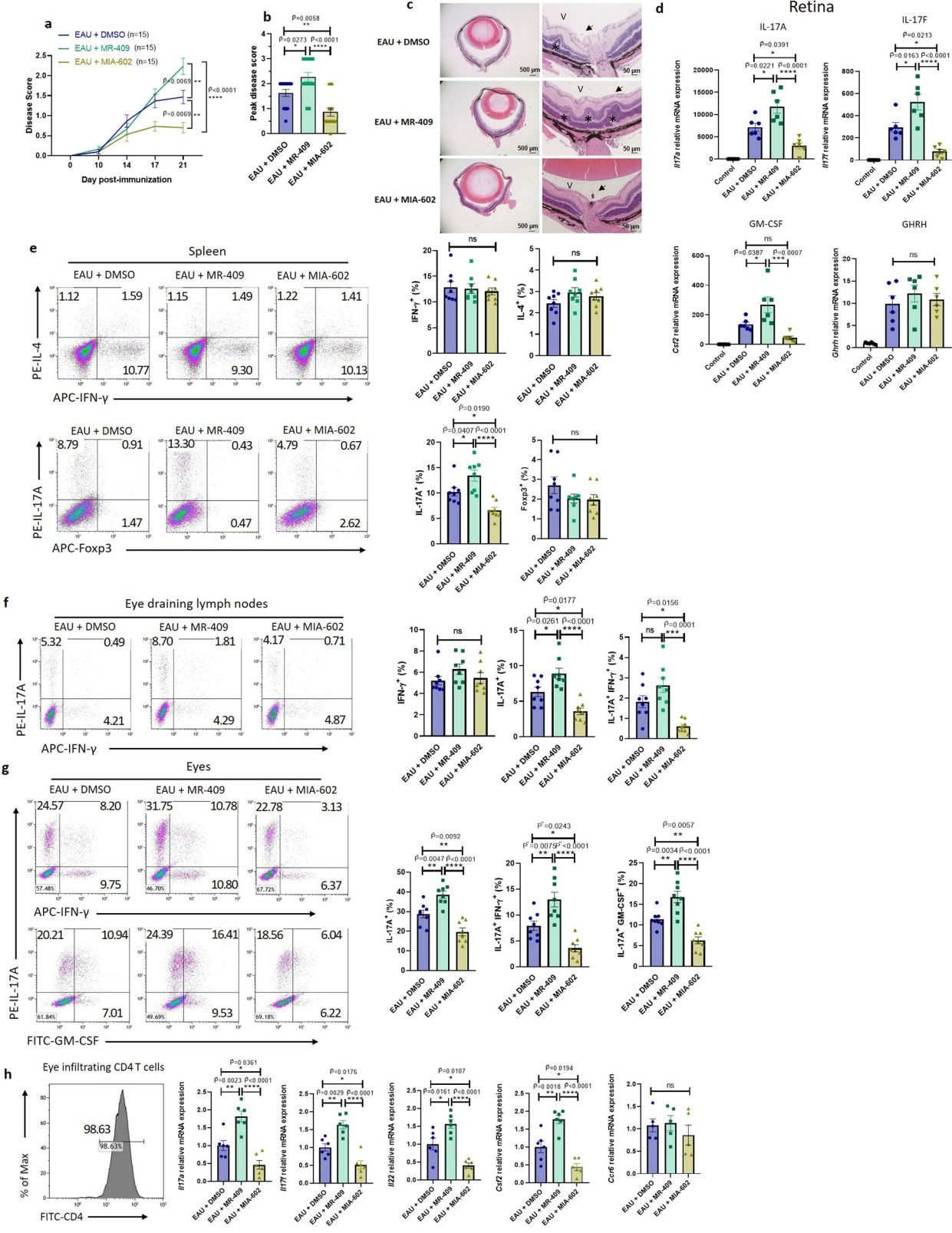

homeostasis, growth, and development caused by GHRH-R deficiency, GHRH agonist and antagonist were treated in WT mice to further verify the effects of enhancing or inhibiting the GHRH-R signaling. Furthermore, to investigate whether GHRH-R signaling acts as a critical regulator in Th17 cell-mediated inflammation, two mouse models of autoimmune diseases, the EAU and EAE models, were employed. Our results showed that GHRH-R signaling is essential for Th17 cell

differentiation and Th17 cell-mediated autoimmune ocular and neural inflammation because the deficiency or inhibition of GHRH-R signaling resulted in (1) reducing non-pathogenic and pathogenic Th17 cell differentiation in vitro; (2) via suppressing the phosphorylation of STAT3 through JAK-STAT3 pathway; (3) impairing the pathogenicity of Th17 cells; and (4) suppressing autoimmune ocular and neural inflammation.

**Fig. 6 | GHRH signaling regulates Th17-mediated autoimmune ocular inflammation.** WT mice were challenged with EAU. From day 10–14 after immunization, mice were treated with DMSO, MR-409 (20 μg/mouse) or MIA-602 (20 μg/mouse) by subcutaneous injection daily to the end point ($n = 15$). **a** Disease score grading of uveitis at five time points after immunization. The *F* value of the one-way ANOVA test is 21.08, and the corresponding *p* value is less than 0.0001 on day 21. **b** Peak disease score of uveitis throughout 21 days after immunization. The *F* value of the one-way ANOVA test is 18.32, and the corresponding *p* value is less than 0.0001. **c** At day 21, eyeballs were collected for H&E staining. Eye-infiltrating cells (arrows) and retinal folds (asterisks) are indicated ($n = 5$). Scale bars: 500 μm (left) and 50 μm (right), respectively. **d** At day 21, retinae were collected for quantifying *Il17a, Il17f, Csf2*, and *Ghrh* expression ($n = 6$). Data were normalized to *Gapdh* and fold change is relative to the non-EAU control. The *F* values of the one-way ANOVA test are 17.50 (*Il17a*), 20.22 (*Il17f*), and 11.53 (*Csf2*), and the corresponding *p* values are 0.0001 (*Il17a*), less than 0.0001 (*Il17f*), and 0.0009 (*Csf2*). The exact *p* values (IL-17A EAU + MR-409 vs EAU + MIA-602, and IL-17F EAU + MR-409 vs EAU + MIA-602) were $8.54 \times 10^{-5}$ and $3.85 \times 10^{-5}$, respectively. **e–g** At day 21, cells were isolated from the spleen, eye-draining lymph nodes, and eyes for flow cytometry analysis. **e** IFN-γ, IL-4, IL-17A and Foxp3 expression in CD4⁺ T cells from spleen ($n = 8$). The *F* value of the one-way ANOVA test is 15.66, and the corresponding *p* value is less than 0.0001. The exact *p* value (IL-17A⁺ EAU + MR-409 vs EAU + MIA-602) was $4.50 \times 10^{-5}$. **f** IFN-γ and IL-17A expression in CD4⁺ T cells from eye-draining lymph nodes ($n = 8$). The *F*

values of the one-way ANOVA test are 17.03 (IL-17A⁺ cells) and 13.74 (IL-17A⁺IFN-γ⁺ cells), and the corresponding *p* values are less than 0.0001 (IL-17A⁺ cells) and 0.0002 (IL-17A⁺IFN-γ⁺ cells). The exact *p* value (IL-17A⁺ EAU + MR-409 vs EAU + MIA-602) was $2.58 \times 10^{-5}$. **g** IL-17A⁺CD4⁺ T cells co-expressing GM-CSF or IFN-γ from eyes ($n = 8$). The *F* values of the one-way ANOVA test are 24.36 (IL-17A⁺ cells), 20.24 (IL-17A⁺IFN-γ⁺ cells), and 26.73 (IL-17A⁺GM-CSF⁺ cells), and the corresponding *p* value are all less than 0.0001. The exact *p* values (IL-17A⁺ EAU + MR-409 vs EAU + MIA-602, IL-17A⁺IFN-γ⁺ EAU + MR-409 vs EAU + MIA-602, and IL-17A⁺GM-CSF⁺ EAU + MR-409 vs EAU + MIA-602) were $2.06 \times 10^{-6}$, $7.99 \times 10^{-6}$, and $1.01 \times 10^{-6}$, respectively. **h** Eye-infiltrating CD4⁺ T cells were enriched and identified by flow cytometry ($n = 10$ per group), followed by quantifying expression of *Il17a, Il17f, Il22, Csf2*, and *Ccr6* (4–6 eyeballs were pooled as one sample and $n = 6$ samples per group). The *F* values of one-way ANOVA test are 24.21 (*Il17a*), 25.87 (*Il17f*), 21.57 (*Il22*), and 27.29 (*Csf2*), and the corresponding *p* value are 0.0002 (*Il17a*), 0.0001 (*Il17f*), 0.0006 (*Il22*), and 0.0001 (*Csf2*). The exact *p* values (*Il17a* EAU + MR-409 vs EAU + MIA-602, *Il17f* EAU + MR-409 vs EAU + MIA-602, *Il22* EAU + MR-409 vs EAU + MIA-602, and *Csf2* EAU + MR-409 vs EAU + MIA-602) were $1.49 \times 10^{5}$, $9.63 \times 10^{-6}$, $2.68 \times 10^{-5}$, and $7.14 \times 10^{-6}$, respectively. Data were the representation of at least two independent experiments. Data were presented as mean ± SEM. *P* values are all determined by one-way ANOVA followed by Bonferroni post hoc test. *, **, ***, and **** represent P̃ < 0.05, P̃ < 0.01, P̃ < 0.001, and P̃ < 0.0001 respectively. ns represents no significant difference.

GHRH-R is barely detectable in naïve T cells, whereas co-stimulatory signals from TCR and CD28 of T cells remarkably induce the expression of the mRNA and protein of GHRH-R as well as the splicing variant GHRH-R SV1. We found that GHRH-R SV1 expression was increased in the activated T cell subset; however, its impact remains unclear. Although the production of GHRH-R on activated T cell does not exhibit a Th cell type-specific pattern, we found that the differentiated Th17 cells have higher GHRH-R expression than other T cell subsets, indicating the synergistic effects of IL-6 or TGF-β in stimulating the expression of GHRH-R. This is similar to IL-23R that is not expressed on naïve T cells, but increasingly expressed on activated T cells upon differentiation into Th17 cells[1,39]. Moreover, there is a strong correlation between GHRH-R and IL-17A production. Th17 cells with higher level of GHRH-R produce more IL-17A. Thus, our results suggest GHRH signaling is not involved in the initial differentiation of Th17 cells but appears to promote the production of IL-17A in activated T cells in the late phase of Th17 differentiation. On the other hand, the differentiation of other T cell subsets, including Th1, Th2, and iTreg, are not affected by the GHRH-R signaling.

Previous studies showed that IL-23R can be induced by IL-6/IL-21 together with low amounts of TGF-β in a RORγt-dependent manner. By responding to IL-23, IL-23R further promotes the synthesis of RORγt in return via enhancing the STAT3 signaling, which is essential for the expansion and stabilization of Th17 cells[4,5,7]. In our study, GHRH-R deficient T cells stimulated by IL-6 plus TGF-β, or a combination of IL-6, IL-1β, and IL-23, showed a reduced expression of IL-23R as well as transcription factors RORγt and RORα, along with Th17 cells related genes *Il17a*, *Il17f*, and *Il22*, whereas the expression of IL-6R, IL-1R, and IL-21R was not altered. These results suggest that GHRH-R signaling is an upstream regulator of RORγt, RORα, and IL-23R. GHRH-R signaling regulated Th17-lineage cytokines through the RORγt-RORα-IL-23 pathway.

Although GHRH-R was expressed in activated T cells, our results indicated that GHRH-R deficiency was able to selectively inhibit the differentiation of Th17 cells without affecting Th1, Th2, and iTreg cells, probably due to the relatively lower expression of GHRH-R in Th1, Th2, and iTreg cells. In addition to the influence in Th17 cell differentiation under the non-pathogenic condition (IL-6 and TGF-β), GHRH-R deficiency reduced IL-17A production in Th17 cells under the pathogenic condition (IL-6, IL-1β, and IL-23). Furthermore, our data showed that even in the absence of exogenous GHRH-R ligands, GHRH-R

expressing T cells stimulated with IL-6 and TGF-β still appeared to have higher IL-17A production than GHRH-R deficient T cells, while the expression of IL-6R, IL-1βR, and IL-21R were not affected, suggesting that GHRH was supplied by T cells in an autocrine/paracrine fashion. However, we could not detect the mRNA of GHRH in differentiated Th17 cells. Previous studies reported that GHRH, GH, and IGF-1 can be secreted by T cells in an autocrine/paracrine fashion to regulate the secretion of cytokines or T cell responses[48–50]. Recently, IGF-1 was found to act as an enhancer to skew the differentiation of Th17 over Treg in autoimmunity[20]. Based on this reported observation, we used the GHRH, GH, or IGF-1 antibodies to neutralize corresponding ligands in the cell culture medium and found that the IL-17A production by differentiated Th17 cells were reduced by the GHRH antibody, but not by the GH or IGF-1 antibodies, indicating that low level of GHRH in the medium had an impact in Th17 cell differentiation. The Th17 cell differentiation could be further enhanced by the GHRH agonist, while it was diminished by the GHRH antagonist or GHRH antibody in vitro. Our results established the impacts of GHRH-R signaling in promoting Th17 cell differentiation.

Given the importance of GHRH-R signaling in Th17 cell differentiation, we showed that the ocular inflammation in the EAU mice was mainly mediated by the GHRH-R signaling. We showed that the expression GHRH and GHRH-R were highly increased in the retina of EAU mouse as well as in eye-infiltrating CD4⁺ T cells, which were consistent with our in vitro results. Furthermore, our previous studies showed that the expression of GHRH and GHRH-R were elevated in the iris and ciliary body of rats challenged with LPS, suggesting GHRH and GHRH-R have an impact in ocular inflammation associated with both innate and adoptive immunity[21,51,52]. However, several questions remain. First, the source of retinal GHRH is not very clear. Although the hypothalamus is thought to be the primary source of GHRH, our results and those of other studies showed that GHRH is expressed in retinal ganglion cells and other retinal cells under physiological and pathological conditions[53,54]. Secondly, the impact of increased GHRH-R expression in the retina during EAU development is unknown. Although the GHRH-GH-IGF axis in the retina is critical for retinal physiology[55,56], our adoptive transfer experiment indicated GHRH-R expression in the retina is not needed for EAU induction. Thirdly, the *Ghrh^{lit/lit}* mice used in this study are whole-body knockout mice instead of T cell-specific knockout mice, we cannot exclude other potential impacts of GHRH-R deficiency on other cell types, which may regulate inflammation in EAU and EAE models in vivo.

Importantly, the deficiency of GHRH-R or the application of the GHRH antagonist significantly alleviated, while GHRH agonist promoted, the clinical severity of EAU. First, the suppressed EAU caused by GHRH-R deficiency or by the GHRH antagonist was associated with a lower frequency of Th17 cells in the spleen, eye-draining lymph nodes, and eyes. In addition, within the eyes, reduced production of GM-CSF or IFN-γ in the CD4⁺IL-17⁺ T cells was in accordance with the inhibited GHRH signaling. In in vitro Th17 cell differentiation, unexpectedly, IFN-γ, GM-CSF, and IL-9 were not detectable in cells, which is different from results obtained from the EAU and EAE mouse models. These discrepancies could be due to multiple rounds of stimulation of Th17 cells to gain pathogenicity in animal models[57]. Alternatively, other potential cytokines produced by innate immune cells may be involved in the development of pathogenic Th17 cells, which cannot be mimicked in in vitro conditions. Furthermore, these declines were associated with downregulated expression of *Il17f*, *Il22*, and *Csf2*, which are the signatures of pathogenic Th17 cells, and were associated with upregulated expression of non-pathogenic Th17 cell-related gene *Il10* in CD4⁺ T cells in the eyes of EAU mice. However, we could not rule out GHRH-R signaling is able to regulate the plasticity of the Th17 cells in EAU mice, although the frequency of IL-17A⁺IFN-γ⁺CD4⁺ T cells in eyes seemed to be regulated by GHRH-R signaling as well, which is different from our in vitro results that GHRH-R signaling did not alter the production of IFN-γ, the marker of Th1 cells, during differentiation. These differences highlighted the complexity of GHRH-R signaling in the differentiation and plasticity of Th17 cells in vivo.

In conclusion, our study demonstrated that GHRH-R signaling acts as an enhancer for Th17 cell differentiation and pathogenicity through the JAK-STAT3 pathway. We also highlighted the critical impacts of GHRH-R signaling in mediating pathogenic Th17 cells in autoimmune ocular and neural inflammation. Our findings shed light on a potential treatment targeting GHRH-R signaling by antagonist or antibodies for Th17 cell-associated autoimmune diseases.

# Methods

## Mice

Wild-type mice (C57BL/6J, Jax 000664) and *Ghrhr^lit/lit* mice (C57BL/6J-*Ghrhr^lit*/J, Jax 000533) were purchased from The Jackson Laboratories. *Ghrhr^lit/lit* mice have a point mutation in both alleles of the *Ghrhr* gene[32,33]. Genotypes were confirmed by quantitative real-time PCR. In this study, both male and female mice, 6–10 weeks old, were used. All animals were kept in a specific pathogen-free facility at 22 °C and 40–50% humidity on a 12/12-h light/dark cycle and fed with standard laboratory chow ad libitum. All animal experiments were conducted in accordance with the guidelines of the Association for Research in Vision and Ophthalmology (ARVO) Statement for the Use of Animals in Ophthalmic and Vision Research. Ethics approval for this study was obtained from the Animal Experimentation Ethics Committee of the Chinese University of Hong Kong (approval numbers 20/014/GRF and 22/270/MIS).

## Induction of EAU and disease assessment

Induction of EAU by active immunization was described previously in ref. 34. Briefly, EAU was induced by subcutaneously immunizing mice with 400 μg IRBP₁₋₂₀ peptide emulsified in an equal volume of complete Freund's adjuvant (CFA) containing 4.5 mg/ml heat-inactivated *M.tuberculosis*. Additionally, mice also received 300 ng *Bordetella pertussis* toxin intraperitoneally on the day of immunization. Mice were euthanized by cervical dislocation. For induction of EAU by adoptive transfer, spleen and lymph nodes cells were harvested from WT or *Ghrhr^lit/lit* donor mice 10 days after immunization and dispersed into single-cell suspension, following enrichment of CD4⁺ T cells by the CD4⁺ T cell isolation kit (STEMCELL). Then these cells were cultured in vitro under Th17 polarizing conditions (50 ng/ml IL-6, 1 ng/ml TGF-β, 10 μg/ml of anti-IFN-γ, and 10 μg/ml of anti-IL4 Abs) and stimulated

with 1 μg/ml of IRBP₁₋₂₀ for 48 h. Then 10 ng/ml IL-23 was treated to cells for a further 24 h. Cells were then purified by centrifugation over Lympholyte-M cell separation media (Cedarlane), washed, and counted. About 5 × 10⁶ cells were injected intraperitoneally into naïve recipient mice. 1 day later, the recipient mice received 200 ng of *Bordetella pertussis* toxin intraperitoneally. Clinical examinations by cSLO, OCT, and ERG were performed on anesthetized mice as previously described in ref. 58. Disease score of EAU was evaluated by fundus examination by the cSLO imaging system (Heidelberg Engineering GmbH) on a scale of 0–4, based on the number, type, and extent of lesions, as described previously in ref. 59. Retinal-choroidal thickness of EAU mice was measured and quantified by OCT imaging system at the same retinal location across various time points (Heidelberg Retina Angiograph 2; Heidelberg Engineering GmbH,) and vision was assessed by scotopic and photopic ERG with a Diagnosys Espion system and the ColorDome light emitting diode (LED) full-field stimulator (Diagnosys LLC)[34].

## Treatment protocols

MR-409 and MIA-602 were synthesized in the laboratory of A.V.S. For in vivo experiments, MR-409 and MIA-602 were dissolved in 100% DMSO for stock and diluted in PBS for injection. The concentration of DMSO was kept below 0.1%. Each mouse was subcutaneously injected with 20 μg MR-409 or MIA-602 dissolved in 100 μL PBS daily on days 10–21 post-immunization. Control mice were injected subcutaneously with 0.1 mL PBS containing 0.1% DMSO. For in vitro experiments, cells were cultured with 100 nM MR-409, hGHRH, or MIA-602 dissolved in 0.01% DMSO for 3 days unless otherwise specified. In some experiments, cells were cultured with ruxolitinib (1 μM) or Stattic (1 μM) for 3 days.

## Induction of EAE and disease assessment

Age and gender-matched mice were immunized with 200 μg of MOG₃₅₋₅₅ peptide emulsified in equal volumes of CFA containing 4.5 mg/ml *M.tuberculosis*. In addition, mice received two doses of 200 ng of *Bordetella pertussis* toxin intraperitoneally on the day of immunization and 2 days after immunization. Disease score of EAE were graded on a scale of 0-5 based on a standard scoring system[60]: 0=no disease, 1=limp tail, 2=hind limb weakness, 3=hind lib paralysis, 4=hind limb and fore limb paralysis, and 5=moribund or death.

## Histology

For EAU mice, on day 21 after immunization, mice were anesthetized and perfused intracardially with PBS, followed by 4% paraformaldehyde (PFA). Eyeballs were immersed in 4% PFA for 24 h and embedded in paraffin. Tissues were sectioned in 5 μm thickness through the pupillary-optic nerve axis for hematoxylin and eosin (H&E) staining. For EAE mice, on day 21 after immunization, mice were anesthetized and perfused intracardially with PBS, followed by 4% PFA. Spinal cords were immersed in 4% PFA for 3–4 days. Tissues were embedded in paraffin and sectioned in 5 μm thickness for H&E or Luxol fast blue staining. All stained slides were captured by a light microscope (DMRB, Leica Microsystems) connected to a Spot digital camera (Diagnostic Instrument).

## Tissue processing

Mice were sacrificed prior to isolation of the spleen, lymph nodes, eyes, or CNS tissues. Secondary lymphoid tissues were collected and grinded into suspension using a syringe and a 70-μm strainer. Red blood cells were lysed using RBC lysis buffer (BioLegend). The remaining cells were washed, filtered with a 40-μm strainer, and resuspended in complete RPMI (RPMI medium containing 10% fetal bovine serum, 100 IU/ml penicillin, 100 μg/ml streptomycin, 1 mM sodium pyruvate, nonessential amino acids, and 55 μM β-mercaptoethanol) for further experiments. For EAU experiments, eyes were collected from mice

challenged with EAU 19–21 days after immunization. After the removal of extraocular tissues, the lens was enucleated from the eyes and the remaining tissue was minced into small pieces with scissors and incubated in a complete RPMI medium containing 1 mg/ml of collagenase D and 1 mg/ml of DNase (QIAGEN) for 45 min at 37 °C. Tissues were then dispersed vigorously by pipetting several times, filtered with a 70-µm strainer, and re-stimulated in the complete RPMI supplemented with 50 ng/ml phorbol myristate acetate (PMA, Sigma), 500 ng/ml ionomycin (Sigma), and 5 µg/ml brefeldin A (BioLegend) for 4 h prior to intracellular staining for flow cytometry. For EAE experiments, spinal cords and brains were collected from mice challenged with EAE 21 days after immunization. These tissues were minced into small pieces and incubated in the complete RPMI containing 1 mg/ml of collagenase D and 1 mg/ml of DNase for 45 min at 37 °C. Then the samples were disrupted by pipetting several times, strained over a 70 µm filter and purified by centrifugation with Percoll gradient at $13,000 \times g$ at 4 °C for 10 min. After washed, cells were resuspended in complete RPMI containing 50 ng/ml PMA (Sigma), 500 ng/ml ionomycin (Sigma), and 5 µg/ml brefeldin A (BioLegend) for 4 h prior to intracellular staining for flow cytometry.

### Flow cytometry

For cell surface staining, the isolated single-cell suspension was incubated in the cell staining buffer (BioLegend) containing anti-CD16/32 antibodies (1:100, BioLegend) for 15 min on ice to block Fc receptors and stained with conjugated fluorescent antibodies of cell surface markers as well as 7-AAD Viability Staining Solution (BioLegend) for 15 min on ice. For intracellular staining for cytokines, cell surface markers and LIVE/DEAD Fixable dye (Thermo Fisher) were stained prior to fixation with 4% PFA in staining buffer for 10 min at room temperature (RT). These cells were washed with intracellular staining permeabilization wash buffer (PB, BioLegend) three times and incubated in PB containing conjugated fluorescent antibodies of intracellular markers for 30 min at room temperature (RT). For intracellular staining of transcription factors, after cell surface marker staining, cells were fixed and permeabilized using True-Nuclear Transcription Factor Buffer Set (BioLegend) or FOXP3 Fix/Perm Buffer Set (BioLegend) for 1 h at RT. Then these cells were washed and stained with conjugated fluorescent antibodies of nuclear transcription factors in the perm buffer from the True-Nuclear Transcription Factor Buffer Set (BioLegend) for 45 min at RT. For intracellular phosphorylated signaling proteins, cell surface marker staining was done prior to fixation in 4% PFA in cell staining buffer for 10 min at RT. Then these cells were permeabilized in ice-cold 90% methanol for 30 min on ice, washed, and incubated with conjugated fluorescent antibodies in cell staining buffer supplemented with cOmplete protease inhibitor cocktail tablets (Roche) for 45 min at RT. All antibodies and buffer sets were listed in Supplementary Data 1. Flow cytometry data was acquired on Cytomics FC500 Flow Cytometry (Beckman), MoFlo Astrios (Beckman), or LSRII (BD Biosciences) and analyzed using Kaluza or FlowJo software.

### In vitro T cell differentiation

Naïve CD4+ (CD62L+CD44−CD25−CD4+) T cells were isolated from spleens and lymph nodes of mice of designated genotypes by negative selection on LS Columns (Miltenyi Biotec) using naive CD4+ T cell isolation kits (Miltenyi Biotec/ STEMCELL) and identified by flow cytometry. Naïve CD4+ T cells ($1 \times 10^5$ cells) were cultured in a 96-well plate pre-coated with 5 µg/ml anti-CD3 antibody (clone 145-2C11, BioLegend) in T cell medium (RPMI medium, 10% fetal bovine serum, 100 IU/ml penicillin, 100 µg/ml streptomycin, 1 mM sodium pyruvate, nonessential amino acids, and 55 µM β-mercaptoethanol) supplemented with 1 µg/ml anti-CD28 (clone 37.51, BioLegend). For Th0 culture, naïve CD4+ T cells were cultured in a T cell medium without any additional cytokines or antibodies. For Th1 cell differentiation, T cells were supplemented with 10 µg/ml of anti-IL-4 (clone 11B11,

BioLegend) and 10 ng/ml of IL-12 (R&D). For Th2 cell differentiation, T cells were supplemented with 10 µg/ml of anti-IFN-γ (clone XMG1.2, BioLegend) and 10 ng/ml of IL-4 (R&D). For non-pathogenic Th17 cell differentiation, T cells were cultured with 10 µg/ml of anti-IL-4 (clone 11B11, BioLegend), 10 µg/ml of anti-IFN-γ (clone XMG1.2, BioLegend), 50 ng/ml of IL-6 (R&D), and 1 ng/ml of TGF-β (R&D). For pathogenic Th17 cell differentiation, T cells were supplemented with 10 µg/ml of anti-IL-4 (clone 11B11, BioLegend), 10 µg/ml of anti-IFN-γ (clone XMG1.2, BioLegend), 50 ng/ml of IL-6 (R&D), 1 ng/ml of TGF-β (R&D) and 10 ng/ml IL-23 (R&D) or combination of cytokines of 50 ng/ml of IL-6 (R&D), 10 ng of IL-1β (R&D) and 10 ng/ml of IL-23(R&D). For iTreg cell differentiation, T cells were supplemented with 10 µg/ml of anti-IL-4 (clone 11B11, BioLegend), 10 µg/ml of anti-IFN-γ (clone XMG1.2, BioLegend), 100 U/ml of IL-2 (Peprotech), and 5 ng/ml of TGF-β (R&D). For in vitro experiments, 100 nM MR-409, hGHRH, and MIA-602 dissolved in DMSO were used unless otherwise specified. T helper cell differentiation was performed under T cell-polarizing conditions for 3 days.

### Cell proliferation assay

Naïve CD4+ T cells were labeled with 1 µM carboxyfluorescein succinimidyl ester (CFSE, BioLegend) following the manufacturer's instruction. Cells were then cultured under Th0 or Th17 cell-polarizing conditions for 3 days before flow cytometry analysis.

### Cell apoptosis assay

Naïve CD4+ T cells were cultured under Th17 cell differentiation condition for 3 or 5 days. Apoptosis was measured by staining cells with propidium iodide (PI, Invitrogen) or 7-AAD viability staining solution (BioLegend) and Annexin V (Invitrogen) following the manufacturer's instructions.

### Immunofluorescence

Naïve CD4+ T cells were isolated and cultured under Th17 differentiation condition for 3 days, washed, and fixed with 4% PFA at RT for 10 min. Cells were then permeabilized in PBS containing 0.2% Triton X-100 and 1% bovine serum albumin (BSA) for 10 min at RT, blocked by 5% BSA in PBS for 30 min and incubated with indicated primary antibodies overnight at 4 °C. After incubation with secondary antibodies and DAPI for 2 h at RT, cells were incubated on poly-ʟ-lysine-coated slides and mounted using GB-Mount (GBI Labs) with coverslips. For retina immunofluorescence, paraffin sections of retina were dewaxed with xylene three times, hydrated, and heated in a pressure cooker for epitope retrieval. Tissues were then permeabilized, blocked, and incubated with primary and secondary antibodies. Subsequently, slides were mounted with coverslips using a DAPI-containing mounting medium. Anti-GHRH Ab (1:100, Abcam, ab187512), anti-GHRH-R Ab (1:100, Abcam, ab76263), FITC-conjugated anti-CD4 Ab (1:50, BioLegend, RM4-5), anti-rabbit secondary Ab (1:100, Alexa Fluor 488/594, Invitrogen), DAPI (1 µg/ml, Invitrogen). Images were captured with a Nikon A1MP confocal microscope using the NIS-Elements software (Nikon). The fluorescence intensity was quantified using ImageJ (National Institutes of Health).

### Isolation of eye-infiltrating CD4+ T cells

EAU mice were sacrificed 19–21 days after immunization and eyes were collected. The lens was enucleated from the eyes and the remaining tissues were minced into small pieces for digestion in a complete RPMI medium containing 1 mg/ml of collagenase D and 1 mg/ml of DNase (QIAGEN) for 45 min at 37 °C. Tissues were then dispersed and filtered with a 70-µm strainer following the centrifugation through a Percoll gradient to acquire mononuclear cells. CD4+ T cells were isolated by negative selection using the Mouse CD4+ T Cell Isolation Kit (STEMCELL). The purity of CD4+ T cells was over 95% by flow cytometry. In this experiment, 5–6 eyes were pooled as one sample for RNA extraction.

## RNA extraction and quantitative real-time PCR

For in vivo experiments, the retina was collected from the eyes of EAU mice 19–21 days after immunization and were homogenized for RNA extraction. For in vitro experiments, naïve CD4$^+$ T cells were cultured under Th17 differentiation conditions for 3 days. Total RNA was extracted from these samples using RNeasy Mini Kit (QIAGEN) following the manufacturer's instructions, quantified, and then converted to cDNA using SuperScript III Reverse Transcriptase (Roche). qRT-PCR was performed using LightCycler 480 Sybr Green I Master (Roche) using LightCycler 480 II real-time PCR (Roche Applied Science). Gene expression was normalized to that of the housekeeping gene *Gapdh* and fold change was calculated by using the $2^{-\Delta\Delta CT}$ threshold cycle method. Relative gene expression was normalized by comparing gene expression to the average of controls that was set to 1. A list of primers is presented in Supplementary Data 1.

## siRNA treatment

GHRH-R expression was knocked down using *Ghrhr* siRNA (siRNA ID254217, Thermo Fisher) by nucleofection. The Silencer Negative Control No. 1 siRNA (AM4611, Thermo Fisher) was used in the control group. Naïve T cells ($1 \times 10^6$ per well) isolated from the spleen and lymph nodes were mixed with 100 μL primary cell nucleofection solution (Lonza). For siRNA transfection, naïve T cells were incubated with 300 nM *Ghrhr* siRNA or Silencer Negative Control siRNA, transferred to Nucleofection cuvette strips (Lonza), and electroporated using a 4D Nucleofector (Lonza) with a DN-100 pulse. After transfection, cells were cultured in 96-well plates with pre-warmed 200 μl complete medium under the Th17 cell differentiation condition for 3 days.

## Statistical analysis

The computer software Prism 8 (GraphPad) was used for statistical analysis and graph plotting. For two-group comparisons, if the data passes the normality test, we use the Student's *t*-test; otherwise, we adopt the Mann–Whitney test for the comparison. When multiple hypotheses are tested, we use the Bonferroni correction to adjust for the multiple comparison and control the family-wise error rate at 0.05. As a result, when *m* hypotheses are tested simultaneously, the ones with the resulting *p* values smaller than *0.05/m* are rejected. For comparisons between three or more groups, we first apply the one-way ANOVA, and we provide the corresponding F statistics and *p* values in the figure legends. When the null hypotheses are rejected, we further compare each pair of groups and show the Bonferroni-adjusted *p* values calculated by Prism 8 on the corresponding figures to account for multiple comparison. The pairwise comparisons with the Bonferroni-adjusted *p* values smaller than 0.05 are significant. To differentiate the unadjusted raw *p* values and the Bonferroni-adjusted *p* values, in the figures, we use "P" to denote the unadjusted raw *p* values and "$\widetilde{P}$" to denote the Bonferroni-adjusted *p* values. All graphs show the mean ± standard error of mean (SEM) and represent three or more samples in at least two independent experiments unless described otherwise.

## Reporting summary

Further information on research design is available in the Nature Portfolio Reporting Summary linked to this article.

## Data availability

All other data are provided in the article and its Supplementary files or from the corresponding author upon request. Source data are deposited in figshare https://doi.org/10.6084/m9.figshare.22739690. Source data are provided with this paper.

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

## Acknowledgements

We thank members of the Chu laboratory for helpful discussions. We thank Professor Kam Tong Leung and Ms Siu Ping Fok for their technical support on the siRNA transfection experiment. We also express our gratitude to Dr Yuzhou Zhang from the Chinese University of Hong Kong, Professor Fengping Shan, and Professor Xun Sun from the China Medical University for their critical comments on this study. This work was supported by the General Research Fund, Research Grants Council, Hong Kong (14104621 and 14102522 to W.K.C., and 14305319 to Y.W.), The Chinese University of Hong Kong Direct Grant (2020.067 and 2021.046 to W.K.C.), National Natural Science Foundation of China (82201198 to J.L.) and Zhejiang Provincial Natural Science Foundation of China (LTGY23H120004 to J.L.). The work of A.V.S. was supported by the Medical Research Service of the US Department of Veterans Affairs.

## Author contributions

Conceptualization, L.D., J.L., and W.K.C. Investigation, L.D., B.M.H., L.Z., Y.W.Y.Y., and J.N.H. Formal analysis, L.D., Y.W., C.C.T., S.O.C., A.V.S., C.P.P., J.L., and W.K.C. Resources, A.V.S. Writing—original draft preparation, L.D., S.O.C., C.P.P., J.L., and W.K.C. Supervision, S.O.C., C.P.P., J.L., and W.K.C. Funding acquisition, Y.W., J.L. and W.K.C. Materials and correspondence requests should be addressed to W.K.C.

## Competing interests

The authors declare no competing interests.
