## [Peer Review File · Nature Communications]

Growth hormone releasing hormone signaling promotes Th17 cell differentiation and autoimmune inflammationREVIEWER COMMENTS

Reviewer #1 (Remarks to the Author):

The manuscript entitled 'Growth hormone releasing hormone signaling promotes Th17 cell differentiation in autoimmune inflammation' by Du et al., focuses on the role of GHRH in promoting Th17 cell differentiation both in vitro and in vivo. The authors provide substantial evidence that GHRH indeed promotes selectively Th17 cell differentiation in vitro, as Th1, Th2 and Treg differentiations were unaffected. GHRH seems to act late in the differentiation process, and to promote the phosphorylation Y705 of STAT3. The results in vivo show similar effects of GHRH in promoting diseases associated with Th17 cells. Although the manuscript is well written and presented, few concerns remain.

1. It seems that since GHRH is a growth factor, GHRH might affect proliferation or survival of Th17 cells directly. Do the authors have any evidence to exclude these possibilities? There is mention of proliferation and apoptosis assays, but the data are provided.
2. In vivo, it is difficult to conclude that GHRH is only acting on th17 cells, since the Ghrhlit/lit mice are whole body KO mice and not T cell dependent. The discussion should reflect this limitation.
3. The effects of GHRH on autoimmune ocular inflammation seems minimal. A discussion of such a difference between the ocular inflammation and the EAE models would be helpful.
4. There are multiple comparisons made, yet there is no correction applied for false discovery rate (figs 2, 4, 5, 6, 7).
5. Some experiments have low number of mice and the statistics are not correct. For example, in Fig 4 and 5, a student t test is not appropriate for n=3 and for comparing multiple comparison groups. F values are not provided when ANOVA are performed. Fig 1E, 6B should be ANOVA, Fig 2 should be Mann Whitney.

Reviewer #2 (Remarks to the Author):

The authors demonstrate that GHRH-R, whose canonical role is in activating the release of GH, is expressed in CD4 T cells and is a key regulator of Th17 differentiation and Th17-driven autoimmune disease affecting the neuroretina. They show that the pathway favors the gene expression signature typical of "pathogenic" Th17 cells and identify the JAK/STAT3 pathway involvement in the mechanism. The authors also demonstrate that GHRH and its receptor are expressed in multiple ocular tissues, but this finding is not followed up, and it is not clear whether or how it relates to the pathology typical of the disease.

Specific comments:

- 1) The Title should be changed to say "...Th17 cell differentiation AND autoimmune inflammation" (instead of IN). These are two independent phenomena: Th17 differentiation is promoted not only in autoimmune disease, but also in the general sense (as shown by the in vitro data).
- 2) The writing is rather verbose and in many places is difficult to follow. Authors describe in detail what can be anyway seen in the Figures, which ends up being distracting to the reader (e.g., including the obvious such as that non-activated cells are not proliferating, e.g., lines 240-241), but sometimes omit significant details that bear on mechanisms and clinical relevance, such as the start day and duration of treatment, e.g., line 275. Some subheadings lack a closing sentence stating the interpretation of the results, e.g., paragraph ending on line 244. The writing needs editing for language and grammar. Incorrect words and sentence structure are sometimes used, making comprehension difficult (e.g., mice "induced with EAU", rats "induced with LPS" – do the authors mean "challenged"?). It is suggested that a professional scientific editing service be engaged to edit the manuscript.

3) Disease is obvious, but in some experiments the scores are quite low, typical of peptide 1–20 of IRBP. Have the authors tried to use the more recently described peptide 651–670?

4) Authors demonstrate increased GHRH and GHRH-R in the inflamed retina and conclude that GHRH signaling plays an important role in response to EAU induction (lines 99-100). They imply a causal relationship, but it may well be a response to inflammation. Reduced eye pathology in (global) GHRH-R deficiency is likely to be a consequence to the reduced Th17 effector response, rather than a consequence of lack of GHRH-R in the retina (lines 115-118). In fact, the adoptive transfer data (fig 2 and fig 3S) show that disease tracks with presence of GHRH-R in the IRBP-specific T cells, not in the retina. These statements need to be rewritten and qualified appropriately.

5) Figure 4: Authors conclude that GHRH-R is only needed for differentiation of the Th17 lineage, from results based on a congenital GHRH-R deficiency. Ideally, these data should be confirmed by GHRH R siRNA knockdown in WT T cells, to exclude developmental effects.

6) The discourse on exogenous vs autocrine GHRH-R deficiency starting in line 196, is unclear. The statements in lines 199-200 and 208-209 seem to directly contradict each other.

7) Fig 7 does not much add over Fig 6 conceptually, and could be moved to supplementary data.

Reviewer #3 (Remarks to the Author):

The authors show results that support the conclusion that GHRH promotes Th17 differentiation and thus Th17-mediated diseases like uveitis and EAE.

The data are generally clean and support the conclusion, but the study remains preliminary for several major reasons.

1. Many panels have text too pixelated to read. Thus the reviewer could not assess some data.

2. N's of 3-4 in many studies, that according to the methods includes both mouse genders and multiple independent experiments (apparently some experiments including an N of a single mouse as written), is entirely too low an N to reach strong conclusions on several points. Assessments like Fig 5A needs more N's for rigorous conclusions about the lack of an effect of anti-GH and anti-IGF1. Work is underpowered to query possible effects of gender, although this is a minor point given the (uncited) literature indicating gender effects of GHRH are not strong.

3. Panels 2A-C should all show same subsets/cytokines- especially the pathogenic IL17A/IFN γ double positive subset. This is especially important for the transfer experiment in the same figure. Same for Fig. 5 where IFN γ , or IL-10, are essential to indicate pathogenicity.

4. Fig. 5B please show scatter plots rather than histograms, which can hide information; text is unreadable.

5. Fig. 5C- unlike other panels, DMSO does not differ from MR 409. Need more Ns

6. Why use 10 days to prime transferred T cells when Fig 1 doesn't show if GHRH/r is up at 10 days? Nor is any cell characterization in earlier panels done at 10 days.

7. What is effect of MR309, MIA602, ruxolitinib or Stattic on cell viability?

8. Please test the GHRH antagonist as a treatment rather than a preventative to add clinical relevance to the basic science slant of the work.

9. Use of t tests is unlikely to be appropriate in panels where N=5-6. Did data pass tests for normality as required for this test to be valid? All stats need checked by a statistician as data shown don't appear to be analyzed as indicated in the methods nor the checklist.

Minor weaknesses:

1. Panel 7E-need IFN γ +IL17a+ frequencies (going along with need for consistent subset/cytokine measures in point 3 above).

Several studies need cited in the intro or discussion as reference limits allow. These include:

(1) Shohreh et al (Endocrinology 2011, 152:3803) connected HGH with increased T cell proliferation and spleen size in a MOG model. And GHRH ko mice were less susceptible to MOG-induced EAE.

(2) Impact of GHRH injection on T cells was published by Khorram J Clin Endocrinol Metab 1997 82:3590, undermining novelty somewhat.

(3) Bodart (Frontiers in Endo, 2018 <https://doi.org/10.3389/fendo.2018.00296>) showed GHRH ko does not impact thymic T cells.

(4) Brown (Mol Therapy 2004 10:644) showed (in cattle) GHRH changes T cell frequencies.

(5) Welniak J Luek Biol 2002 71:381 reviewed T cells and growth hormone-references therein may be useful.

These findings need to be highlighted as background for the "next logical step" study presented or to fit outcomes into the context of the literature in the discussion, which at present is largely a repeat of the results and needs extensive editing.

Reviewer #1 (Remarks to the Author):

The manuscript entitled ‘Growth hormone releasing hormone signaling promotes Th17 cell differentiation in autoimmune inflammation’ by Du et al., focuses on the role of GHRH in promoting Th17 cell differentiation both in vitro and in vivo. The authors provide substantial evidence that GHRH indeed promotes selectively Th17 cell differentiation in vitro, as Th1, Th2 and Treg differentiations were unaffected. GHRH seems to act late in the differentiation process, and to promote the phosphorylation Y705 of STAT3. The results in vivo show similar effects of GHRH in promoting diseases associated with Th17 cells. Although the manuscript is well written and presented, few concerns remain.

1. It seems that since GHRH is a growth factor, GHRH might affect proliferation or survival of Th17 cells directly. Do the authors have any evidence to exclude these possibilities? There is mention of proliferation and apoptosis assays, but the data are provided.

RE: Thank you very much for the comments. We have performed additional experiments (new Fig. S5F and line 259) in which we checked the Ki67 level, a cellular marker detected in cycling cells. Both WT and *Ghrhr*^{lit/lit} Th17 cells expressed similar Ki67 levels, indicating that GHRH signaling does not impact Th17 cell proliferation. Also, we checked the Annexin V and 7-AAD cell viability levels (new Fig. S5G and line 259). Both WT and *Ghrhr*^{lit/lit} Th17 cells, in the presence of GHRH antagonist MIA-602 or GHRH agonist MR-409, showed no significant differences in the frequency of Annexin V⁺ 7AAD⁻ cells, indicating GHRH signaling does not impact Th17 cell viability.

2. In vivo, it is difficult to conclude that GHRH is only acting on th17 cells, since the *Ghrhlit/lit* mice are whole body KO mice and not T cell dependent. The discussion should reflect this limitation.

RE: Thank you very much for this important comment. In our study, *Ghrhr*^{lit/lit} mice produced less ocular inflammation and fewer Th17 cells in EAU and EAE models *in vivo*. Additionally, naïve T cells from *Ghrhr*^{lit/lit} mice produced less IL-17 under Th17 cell differentiation conditions. Because *Ghrhr*^{lit/lit} mice are whole-body knockout mice instead of T cell-specific knockout mice, we cannot exclude other potential impacts of GHRH-R deficiency on other cell types, which may regulate inflammation in EAU and EAE models *in vivo*. We have reflected this limitation in the discussion (line 416-419).

3. The effects of GHRH on autoimmune ocular inflammation seems minimal. A discussion of such a difference between the ocular inflammation and the EAE models would be helpful.

RE: In our study, previously we only used the typical IRBP₁₋₂₀ peptide for immunization. Because mouse strain C57BL/6 shows a moderate susceptibility to disease by immunization with IRBP₁₋₂₀ peptide, in our case, a higher dose of IRBP₁₋₂₀ peptide (400 µg per mouse) in complete Freund’s adjuvant (CFA) was injected subcutaneously with intraperitoneal injection of pertussis toxin (PTX, 0.3 µg) for immunization. Furthermore, we developed a more sensitive scoring system based on cSLO, Oct and ERG (Li J, et al. *Investigative Ophthalmology & Visual Science*, 2017;58(10):4193-4200), which assessed disease severity using quantified retinal thickness and visual function, to validate our finding that GHRH-R deficiency protected animals immunized by IRBP₁₋₂₀ from EAU. And as suggested by Reviewer 2, we performed additional experiments, in

which WT and *Ghrhr*^{lit/lit} mice (8 mice per group) were immunized with IRBP₆₅₁₋₆₇₀, CFA and PTX. Mice immunized with IRBP₆₅₁₋₆₇₀ showed more severe clinical manifestations of EAU. Consistently, compared with WT mice, *Ghrhr*^{lit/lit} mice immunized with IRBP₆₅₁₋₆₇₀ developed less EAU in terms of disease score and quantified fold change of retinal choroidal thickness (new Fig. S1 E-H and line 126-129). EAU induced by the immunization of IRBP₆₅₁₋₆₇₀ showed similar clinical scores with the EAE model.

4. There are multiple comparisons made, yet there is no correction applied for false discovery rate (figs 2, 4, 5, 6, 7).

RE: Thank you very much for the comment. In Fig. 2F, 4F, 5A, B and E to G, 6A, B and D to H, P values were determined by one-way ANOVA with Bonferroni correction for multiple comparisons. In Fig. 7A and C (now move to Fig. S7A and S7C), P values were determined by Mann-Whitney test in two group comparison on day 21.

5. Some experiments have low number of mice and the statistics are not correct. For example, in Fig 4 and 5, a student t test is not appropriate for n=3 and for comparing multiple comparison groups. F values are not provided when ANOVA are performed. Fig 1E, 6B should be ANOVA, Fig 2 should be Mann Whitney.

RE: Thank you very much. In Fig. 4 and 5, we have worked on additional animals to increase the n to 6 mice per group. For multiple comparisons, P value was determined by one-way ANOVA with Bonferroni correction. F values have been added to the legends of Fig 2F, 3A and D, 4F, 5A, B, E to G, and 6A, B, D to H. For two-group comparison, P value was determined by Mann-Whitney test.

In Fig. 1E, only two groups were compared at various time points. At the specific time points (Days 10, 14, 17 and 21), the data does not follow a normal distribution. Therefore, P values were determined by Mann-Whitney test at that specific time points. In Fig. 6B, the P value is determined by one-way ANOVA with Bonferroni correction for multiple comparison. In Fig. 2A to D, P values are determined by Mann-Whitney test for two group comparison. In Fig. 2F, at various specific time points, P value was determined by one-way ANOVA with Bonferroni correction for multiple comparison.

Reviewer #2 (Remarks to the Author):

The authors demonstrate that GHRH-R, whose canonical role is in activating the release of GH, is expressed in CD4 T cells and is a key regulator of Th17 differentiation and Th17-driven autoimmune disease affecting the neuroretina. They show that the pathway favors the gene expression signature typical of “pathogenic” Th17 cells and identify the JAK/STAT3 pathway involvement in the mechanism. The authors also demonstrate that GHRH and its receptor are expressed in multiple ocular tissues, but this finding is not followed up, and it is not clear whether or how it relates to the pathology typical of the disease.

Specific comments:

1. The Title should be changed to say "...Th17 cell differentiation AND autoimmune inflammation" (instead of IN). These are two independent phenomena: Th17 differentiation is promoted not only in autoimmune disease, but also in the general sense (as shown by the in vitro data).

RE: Thank you very much. We have updated the title into "Growth hormone releasing hormone signaling promotes Th17 cell differentiation and autoimmune inflammation."

2. The writing is rather verbose and in many places is difficult to follow. Authors describe in detail what can be anyway seen in the Figures, which ends up being distracting to the reader (e.g., including the obvious such as that non-activated cells are not proliferating, e.g., lines 240-241), but sometimes omit significant details that bear on mechanisms and clinical relevance, such as the start day and duration of treatment, e.g., line 275. Some subheadings lack a closing sentence stating the interpretation of the results, e.g., paragraph ending on line 244. The writing needs editing for language and grammar. Incorrect words and sentence structure are sometimes used, making comprehension difficult (e.g., mice "induced with EAU", rats "induced with LPS" – do the authors mean "challenged"?). It is suggested that a professional scientific editing service be engaged to edit the manuscript.

RE: Thank you very much for correcting our writing style. We have simplified the sentence on line 257-258. And we have added the treatment start day and duration on line 293. We have also added a concluding statement on line 261-262. The language and wordings have been refined on line 114, 319, 404, 516, 524, 832, 844, 921, 948 and 989.

3. Disease is obvious, but in some experiments the scores are quite low, typical of peptide 1–20 of IRBP. Have the authors tried to use the more recently described peptide 651–670?

RE: Thank you very much for the insightful comments. In our study, we only used the typical IRBP₁₋₂₀ peptide for immunization. Because mouse strain C57BL/6 shows a moderate susceptibility to disease by immunization with IRBP₁₋₂₀ peptide, in our case, a higher dose of IRBP₁₋₂₀ peptide (400 µg per mouse) in complete Freund's adjuvant (CFA) was injected subcutaneously with intraperitoneal injection of pertussis toxin (PTX, 0.3 µg) for immunization. Furthermore, we developed a more sensitive scoring system based on cSLO, Oct and ERG (*Li J, et al. Investigative Ophthalmology & Visual Science, 2017;58(10):4193-4200*), which assessed disease severity using quantified retinal thickness and visual function, to validate our finding that GHRH-R deficiency protected animals immunized by IRBP₁₋₂₀ from EAU. And as suggested, we performed additional experiments, in which WT and *Ghrhr*^{lit/lit} mice (8 mice per group) were immunized with IRBP₆₅₁₋₆₇₀, CFA and PTX. Mice immunized with IRBP₆₅₁₋₆₇₀ showed more severe clinical manifestations of EAU. Consistently, compared with WT mice, *Ghrhr*^{lit/lit} mice immunized with IRBP₆₅₁₋₆₇₀ developed less EAU in terms of disease score and quantified fold change of retinal choroidal thickness (new Fig. S1 E-H and line 126-129).

4. Authors demonstrate increased GHRH and GHRH-R in the inflamed retina and conclude that GHRH signaling plays an important role in response to EAU induction (lines 99-100). They imply a causal relationship, but it may well be a response to inflammation. Reduced eye pathology in (global) GHRH-R deficiency is likely to be a consequence to the reduced Th17 effector response, rather than a consequence of lack of GHRH-R in the retina (lines 115-118). In fact, the adoptive

transfer data (fig 2 and fig 3S) show that disease tracks with presence of GHRH-R in the IRBP-specific T cells, not in the retina. These statements need to be rewritten and qualified appropriately.

RE: We thank the reviewer for this excellent comment. We agree that the enhanced expression of GHRH and GHRH-R in the retina could be a response to inflammation. And we agree that the reduced eye pathology is likely due to the lack of GHRH-R in Th17 cells, rather than in retina. We have rewritten the summary statements on line 108-109 and 129-131.

5. Figure 4: Authors conclude that GHRH-R is only needed for differentiation of the Th17 lineage, from results based on a congenital GHRH-R deficiency. Ideally, these data should be confirmed by GHRH R siRNA knockdown in WT T cells, to exclude developmental effects.

RE: Thank you very much. The underlying developmental effects of congenital GHRH-R deficiency on Th17 cell is also our concern. We attempted to exclude the development effects by applying the GHRH agonist and antagonist in *in vitro* Th17 cell differentiation to validate our findings from a pharmacological angle (Fig. 5B).

Additionally, we have performed experiments by knocking down GHRH-R expression with siRNA. WT naïve T cells isolated from spleen and lymph nodes were incubated with *Ghrhr* siRNA or control siRNA. After transfection, naïve T cells were cultured under the Th17 cell differentiation condition for 3 days. Our result demonstrated that in the *Ghrhr* siRNA treated group, the IL-17A production in CD4⁺ T cells was significantly lower, along with lower GHRH-R expression, compared with the control siRNA treated group (new Fig. S5K and line 248-250).

6. The discourse on exogenous vs autocrine GHRH-R deficiency starting in line 196, is unclear. The statements in lines 199-200 and 208-209 seem to directly contradict each other.

RE: Thank you very much. We have avoided the descriptions of exogenous and autocrine GHRH and now we have clarified the statement on line 211-212. We could not detect the expression of *Ghrh* in Th17 cells under the stimulation of IL-6 with TGF- β or IL-6, IL-1 β with IL-23. However, antibody against GHRH could significantly decreased the expression of IL-17A in WT Th17 cells, which we could only conclude GHRH in the cell culture environment could influence the differentiation of Th17 cells. We have updated the statement on line 223-224 to avoid the contradiction.

7. Fig 7 does not much add over Fig 6 conceptually, and could be moved to supplementary data.

RE: Thank you very much. We have moved Fig 7 to supplementary data (Fig. S7).

Reviewer #3 (Remarks to the Author):

The authors show results that support the conclusion that GHRH promotes Th17 differentiation and thus Th17-mediated diseases like uveitis and EAE.

The data are generally clean and support the conclusion, but the study remains preliminary for several major reasons.

1. Many panels have text too pixelated to read. Thus the reviewer could not assess some data.
RE: We are very sorry for the pixelated figures within the system generated PDF file. We have uploaded higher resolution figures to the online submission system.

2. N's of 3-4 in many studies, that according to the methods includes both mouse genders and multiple independent experiments (apparently some experiments including an N of a single mouse as written), is entirely too low an N to reach strong conclusions on several points. Assessments like Fig 5A needs more N's for rigorous conclusions about the lack of an effect of anti-GH and anti-IGF1. Work is underpowered to query possible effects of gender, although this is a minor point given the (uncited) literature indicating gender effects of GHRH are not strong.

RE: Thank you very much for the comments. We have collected additional data from more animals in Fig 5A, B, D, E, F and G. We agree the gender effects of GHRH are not strong. According to a published paper (*Kamegai J, et al. Journal of Neuroendocrinology, 1999; 11:299-306*), GHRH-R mRNA levels were comparable in male and female rats.

3. Panels 2A-C should all show same subsets/cytokines- especially the pathogenic IL17A/IFNg double positive subset. This is especially important for the transfer experiment in the same figure. Same for Fig. 5 where IFNg, or IL-10, are essential to indicate pathogenicity.

RE: Thank you very much. As suggested, the data in Fig. 2A-C are now further elaborated in the consistent subset of IFN- γ ⁺ IL-17⁺ as in Fig. 2B-C and Fig. S3I. To determine pathogenicity, we performed new experiments: naïve T cells from WT and *Ghrhr*^{lit/lit} mice were cultured under pathogenic (IL-6 and TGF- β) or non-pathogenic (IL-6, TGF- β and IL-23) Th17 cell-polarizing conditions for 3 days in the presence of MR-409 or MIA-602. The subsets IL-17⁺ and IFN- γ ⁺ IL-17⁺ in CD4⁺ T cells were analyzed (new Fig. S5L-M and line 241-243).

4. Fig. 5B please show scatter plots rather than histograms, which can hide information; text is unreadable.

RE: Thank you very much. In Fig. 5B, the data has been already showed in scatter plots. In addition, we have added the scatter plot in Fig. 5D.

5. Fig. 5C- unlike other panels, DMSO does not differ from MR 409. Need more Ns

RE: Thank you very much. We did not treat DMSO in Fig. 5C. On the other hand, we did treat DMSO in Fig. 5B. And now we have added more Ns in Fig. 5B. Our new data in Fig. 5B showed that WT cells treated with MR-409 have higher IL-17A production than WT cells treated with DMSO.

6. Why use 10 days to prime transferred T cells when Fig 1 doesn't show if GHRH/r is up at 10 days? Nor is any cell characterization in earlier panels done at 10 days.

RE: Thank you very much for this important comment. In our study, for EAU induction in mice by adoptive transfer, IRBP-specific CD4⁺ T cells were isolated from the spleen and lymph nodes

of donor mice 10 days after immunization. Then these cells were cultured *in vitro* under pathogenic Th17 polarizing conditions in stimulation with IRBP for 3 days for adoptive transfer into naïve recipient mice. There are two reasons why donor mice were immunized with IRBP for 10 days, and then the enriched donor CD4⁺ T cells were restimulated by IRBP under pathogenic Th17 cell polarizing conditions for another 3 days. Firstly, for EAU induction in mice by active immunization, the ocular inflammation started to appear at days 9-11 (Fig. 1E), when the IRBP-specific effector T cells had already been primed and was able to enter into the eyes. Secondly, in our *in vitro* experiments, the gene expression of *Ghrhr* was significantly increased in CD4⁺ T cell after stimulation with anti-CD3 antibodies under Th17 cell-polarizing condition for 48-72h (Fig. 3A and B). Therefore, after IRBP-specific CD4⁺ T cells were isolated and enriched from spleen and lymph nodes of donor mice, and were restimulated with IRBP *in vitro* under pathogenic Th17 polarizing conditions for 72 h, which could maintain the expression of GHRH-R at a high level. By using this protocol, we saw that recipient mice who received T cells from WT mice, rather than *Ghrhr*^{lit/lit} mice, were able to develop ocular inflammation.

7. What is effect of MR409, MIA602, ruxolitinib or Stattic on cell viability?

RE: Thank you very much for this comment. To determine whether MR-409, MIA-602, ruxolitinib and stattic influenced cell viability, we employed the Annexin V and 7-AAD cell viability assay. Our results (new Fig. S5G and line 259) showed that neither MIA-602 nor MR-409 had any detectable effect on cell viability (Annexin V⁻ 7AAD⁻) in WT and *Ghrhr*^{lit/lit} T cells. Furthermore, WT and *Ghrhr*^{lit/lit} T cells treated with ruxolitinib or stattic had significantly lower cell viability than cells treated with the DMSO control. However, there is no significant difference in cell viability between WT and *Ghrhr*^{lit/lit} cells treated with ruxolitinib or stattic (new Fig. S5G and line 259).

8. Please test the GHRH antagonist as a treatment rather than a preventative to add clinical relevance to the basic science slant of the work.

RE: Thank you very much for this important comment. In our EAU model, the ocular inflammation occurred on days 9-11 (Fig. 1E). And mice were treated with GHRH antagonist MIA-602 daily by subcutaneous injection on days 10-21 after immunization. Therefore, MIA-602 was applied as a treatment rather than a preventative. We have clarified our treatment protocol in the method (line 293).

9. Use of t tests is unlikely to be appropriate in panels where N=5-6. Did data pass tests for normality as required for this test to be valid? All stats need checked by a statistician as data shown don't appear to be analyzed as indicated in the methods nor the checklist.

RE: Thank you very much. As suggested by Reviewer 1 and 2, we have worked on additional animals to increase the n to 6 animals per group in Fig. 4 and 5. In this study, T tests were applied only if the data passed normality test. For non-parametric data, P values were determined by Mann-Whitney test.

Minor weaknesses:

1. Panel 7E-need IFN γ +IL17a+ frequencies (going along with need for consistent subset/cytokine measures in point 3 above).

RE: Thank you very much. Fig. 7 has been moved to supplementary as suggested by Reviewer 2. Fig. S7E is now shown in the consistent subset (IFN- γ ⁺, IL-17⁺ and IFN- γ ⁺ IL-17⁺).

Several studies need cited in the intro or discussion as reference limits allow. These include: (1) Shohreh et al (Endocrinology 2011, 152:3803) connected HGH with increased T cell proliferation and spleen size in a MOG model. And GHRH ko mice were less susceptible to MOG-induced EAE.

(2) Impact of GHRH injection on T cells was published by Khorram J Clin Endocrinol Metab 1997 82:3590, undermining novelty somewhat.

(3) Bodart (Frontiers in Endo, 2018 <https://doi.org/10.3389/fendo.2018.00296>) showed GHRH ko does not impact thymic T cells.

(4) Brown (Mol Therapy 2004 10:644) showed (in cattle) GHRH changes T cell frequencies.

(5) Welniak J Luek Biol 2002 71:381 reviewed T cells and growth hormone-references therein may be useful.

These findings need to be highlighted as background for the “next logical step” study presented or to fit outcomes into the context of the literature in the discussion, which at present is largely a repeat of the results and needs extensive editing.

RE: Thank you very much. Now we have revised the introduction to highlight the findings of these references (line 75-83 and 218).

REVIEWER COMMENTS

Reviewer #1 (Remarks to the Author):

The authors have responded to most of my comments except for the statistics. The statistics are still not correct for all the figures, and the figure legends do not match the material and methods in term of statistics used. These need to be fixed.

Reviewer #2 (Remarks to the Author):

The authors have devoted effort to address the reviewers' concerns thoroughly, both experimentally by providing new data and through changes in the text. The revised manuscript is considerably improved and provides novel understanding of the regulation of the Th17 response. My original comments have been addressed and I have no further concerns.

Reviewer #3 (Remarks to the Author):

Lingering comments/improvements/concerns:

Line 80 "In contrast" needs to say something more like "in agreement with these results". In many of the figures font remains extremely small, even at 200% on screen. So do many of the fonts in the supplement, which won't be readable upon printing.

Fig 1C needs quantification.

Flow panels that have compensation problems that need resolved for more rigor: S1A, S3A, S3I, 2C, 6E, 6G; Fig 2B difference in dot plots is not convincing- collecting more events might mitigate this concern. Adding more Th17 markers to RNA analysis might solve this problem if flow antibodies aren't cooperating.

Fig 3 C- The scale bar is not detectable to confirm both cell types are at the same magnification.

Line 197 "inhibited" is an active process. Data don't show this-they show signal is lower.

Some panels of Fig. 3+4, Fig. 6 N's remain low (N=3-4) despite claim of "at least two independent experiments".

If these pathogenic Th17 cells don't make IFN γ in polarization studies (line 242) what makes them pathogenic? Is there another cytokine that is differentiating these from non-pathogenic cells? What are IL-10 and IL-9 levels? Fig. 6 makes the opposite point-that IFN γ + Th17s are pathogenic. How to unify these findings?

Fig 5F-putting Y axis on same scale would help emphasize differences better.

Discussion largely repeats results and needs edited extensively. Text like that starting at line 408-422 are more appropriate discussion material.

Reviewer #1 (Remarks to the Author):

The authors have responded to most of my comments except for the statistics. The statistics are still not correct for all the figures, and the figure legends do not match the material and methods in term of statistics used. These need to be fixed.

RE: Thank you very much for the comments. In the revised version, now we have elaborated the statistical methods in both the material and methods section (line 640-654) and the figure legends to clarify potential confusion.

Specifically, for two-group comparisons, if the data passed the normality test, we use the Student's t-test; otherwise, we adopt the Mann-Whitney test for the comparison. When multiple hypotheses are tested, we use the Bonferroni correction to adjust for the multiple comparison. Among all the methods for multiple testing correction, which include the false discovery rate approach, the Bonferroni correction is the most conservative one and is applicable even when there exists dependence between the test statistics for the multiple hypotheses under testing. Therefore, we adopt the Bonferroni correction in this manuscript and control the family-wise error rate at 0.05. As a result, when m hypotheses are tested simultaneously, the ones with the resulting p-values smaller than $0.05/m$ are rejected.

For comparisons between three or more groups, we first apply the one-way ANOVA, and we provide the corresponding F statistics and p-values in the figure legends. When the null hypotheses are rejected, we further compare each pair of groups and show the Bonferroni-adjusted p-values calculated by the computer software Prism 8 (Graphpad) on the corresponding figures to account for multiple comparison. The pair-wise comparisons with the Bonferroni-adjusted p-values smaller than 0.05 are rejected. To differentiate the unadjusted raw p-values and the Bonferroni-adjusted p-values, in the figures, we use "P" to denote the unadjusted raw p-values and " \tilde{P} " to denote the Bonferroni-adjusted p-values.

We hope the elaboration help to clarify the potential confusion.

Reviewer #2 (Remarks to the Author):

The authors have devoted effort to address the reviewers' concerns thoroughly, both experimentally by providing new data and through changes in the text. The revised manuscript is considerably improved and provides novel understanding of the regulation of the Th17 response. My original comments have been addressed and I have no further concerns.

RE: Thank you very much for the comments. Our manuscript has been improved a lot with your comments.

Reviewer #3 (Remarks to the Author):

Lingering comments/improvements/concerns:

1. Line 80 "In contrast" needs to say something more like "in agreement with these results".

RE: Thank you very much. We have changed “In contrast” into “In agreement with these results” on line 80.

2. In many of the figures font remains extremely small, even at 200% on screen. So do many of the fonts in the supplement, which won't be readable upon printing.

RE: Thank you very much for the comments. We are very sorry for the pixelated figures within the system-generated PDF file. We have uploaded higher resolution figures to the online submission system. In addition, we have enlarged the fonts in most figures and added additional labels in every flow cytometry figure.

3. Fig 1C needs quantification.

RE: Thank you very much. We have added the fluorescence intensity quantification analysis in new Fig. 1C

4. Flow panels that have compensation problems that need resolved for more rigor: S1A, S3A, S3I, 2C, 6E, 6G; Fig 2B difference in dot plots is not convincing- collecting more events might mitigate this concern. Adding more Th17 markers to RNA analysis might solve this problem if flow antibodies aren't cooperating.

RE: Thank you very much for the comments. We did a rigorous check on the compensation in flow cytometry. Figs. S3A and S3I were adjusted (new Figs. S3A and S3I) based on the adjusted compensation. This update did not affect our conclusions. For Figs S2A, 2C, 6E and 6G, we have tried our best to fine tune the compensation settings. Now there are only minor spillovers of fluorescence signals, which did not affect our conclusions. Fig. 2B has been replaced with a new scatter plot with more events collected (new Fig. 2B).

5. Fig 3 C- The scale bar is not detectable to confirm both cell types are at the same magnification.

RE: Thank you very much. The scale bars on both cell types are at the same magnification. In addition, the font size of scale bar has been enlarged (new Fig 3C).

6. Line 197 “inhibited” is an active process. Data don't show this-they show signal is lower.

RE: Thank you very much. We have changed “inhibited” into “had a reduced capacity” on line 197-199.

7. Some panels of Fig. 3+4, Fig. 6 N's remain low (N=3-4) despite claim of “at least two independent experiments”.

RE: Thank you very much. In Figs. 2D, 3A, 3D, 4B, 4D, 4G, 5C, 6D, 6G and 6H, we have combined data from two independent experiments and increased N's to 6-8 per group.

8. If these pathogenic Th17 cells don't make IFN γ in polarization studies (line 242) what makes them pathogenic? Is there another cytokine that is differentiating these from non-pathogenic cells? What are IL-10 and IL-9 levels? Fig. 6 makes the opposite point-that IFN γ + Th17s are pathogenic. How to unify these findings?

RE: Thank you very much for the comments. In GHRH-R-deficient animals or cells, higher levels of IL-10, an anti-inflammatory cytokine, and lower levels of IL-22, a pathogenic Th17 cytokine that plays an important role in many autoimmune diseases, were detected, indicating that GHRH-R deficiency impaired Th17 cell pathogenicity (Figs. 2D and 5C). Furthermore, as suggested, we have performed new experiments. WT and GHRH-R-deficient naïve T cells were cultured for 3 days under Th17 cell *in vitro* differentiation conditions (IL-6/TGF- β or IL-6/TGF- β /IL-23), and the gene expression of IL-10 was examined. We found that GHRH-R-deficient cells expressed a higher IL-10 level than WT cells, indicating that GHRH-R deficiency reduces the pathogenicity of Th17 cells by increasing IL-10 production (new Fig. S5N and line 242-245). Unfortunately, the gene expression of IL-9 was not detectable under *in vitro* Th17 cell differentiation conditions. The different expression levels of IFN- γ , GM-CSF and IL-9 between *in vitro* differentiation and in EAU and EAE mouse models could be due to multiple rounds of stimulation of Th17 cells to gain pathogenicity in animal models (Hirota et al. Nature Immunology12, p255, 2011). Alternatively, other potential cytokines produced by innate immune cells may be involved in the development of pathogenic Th17 cells, which cannot be mimicked in *in vitro* conditions. We have reflected this point in the discussion (line 424-429).

9. Fig 5F-putting Y axis on same scale would help emphasize differences better.

RE: Thank you very much. The Y axis has been revised to be on the same scale (new Fig 5F).

10. Discussion largely repeats results and needs edited extensively. Text like that starting at line 408-422 are more appropriate discussion material.

RE: Thank you very much for the comments. The discussion has been simplified and the repeated contents have been removed (line 408-415).

REVIEWERS' COMMENTS

Reviewer #1 (Remarks to the Author):

The authors have answered all my comments.

Reviewer #2 Communicated directly to the editor that all their concerns were addressed.

Reviewer #1:

The authors have answered all my comments.

RE: Thank you very much.

Reviewer #2:

Communicated directly to the editor that all their concerns were addressed.

RE: Thank you very much.